# Bootstrapped Meta-Learning

**Sebastian Flennerhag**
DeepMind
flennerhag@google.com

**Yannick Schroecker**
DeepMind

**Tom Zahavy**
DeepMind

**Hado van Hasselt**
DeepMind

**David Silver**
DeepMind

**Satinder Singh**
DeepMind

## ABSTRACT

Meta-learning empowers artificial intelligence to increase its efficiency by learning how to learn. Unlocking this potential involves overcoming a challenging meta-optimisation problem. We propose an algorithm that tackles this problem by letting the meta-learner teach itself. The algorithm first bootstraps a *target* from the meta-learner, then optimises the meta-learner by minimising the *distance* to that target under a chosen (pseudo-)metric. Focusing on meta-learning with gradients, we establish conditions that guarantee performance improvements and show that the metric can control meta-optimisation. Meanwhile, the bootstrapping mechanism can extend the effective meta-learning horizon without requiring backpropagation through all updates. We achieve a new state-of-the art for model-free agents on the Atari ALE benchmark and demonstrate that it yields both performance and efficiency gains in multi-task meta-learning. Finally, we explore how bootstrapping opens up new possibilities and find that it can meta-learn efficient exploration in an $\varepsilon$-greedy $Q$-learning agent—without backpropagating through the update rule.

## 1 INTRODUCTION

In a standard machine learning problem, a *learner* or *agent* learns a task by iteratively adjusting its parameters under a given update rule, such as Stochastic Gradient Descent (SGD). Typically, the learner's update rule must be tuned manually. In contrast, humans learn seamlessly by relying on previous experiences to inform their learning processes (Spelke & Kinzler, 2007).

For a (machine) learner to have the same capability, it must be able to learn its update rule (or such inductive biases). *Meta-learning* is one approach that learns (parts of) an update rule by applying it for some number of steps and then evaluating the resulting performance (Schmidhuber, 1987; Hinton & Plaut, 1987; Bengio et al., 1991). For instance, a well-studied and often successful approach is to tune parameters of a gradient-based update, either online during training on a single task (Bengio, 2000; Maclaurin et al., 2015; Xu et al., 2018; Zahavy et al., 2020), or meta-learned over a distribution of tasks (Finn et al., 2017; Rusu et al., 2019; Flennerhag et al., 2020; Jerfel et al., 2019; Denevi et al., 2019). More generally, the update rule can be an arbitrary parameterised function (Hochreiter et al., 2001; Andrychowicz et al., 2016; Kirsch et al., 2019; Oh et al., 2020), or the function itself can be meta-learned jointly with its parameters (Alet et al., 2020; Real et al., 2020).

Meta-learning is challenging because to evaluate an update rule, it must first be applied. This often leads to high computational costs. As a result most works optimise performance after $K$ applications of the update rule and assume that this yields improved performance for the remainder of the learner's lifetime (Bengio et al., 1991; Maclaurin et al., 2015; Metz et al., 2019). When this assumption fails, meta-learning suffers from a short-horizon bias (Wu et al., 2018; Metz et al., 2019). Similarly, optimizing the learner's performance after $K$ updates can fail to account for the *process* of learning, causing another form of myopia (Flennerhag et al., 2019; Stadie et al., 2018; Chen et al., 2016; Cao et al., 2019). Challenges in meta-optimisation have been observed to cause degraded lifetime performance (Lv et al., 2017; Wichrowska et al., 2017), collapsed exploration (Stadie et al., 2018; Chen et al., 2016), biased learner updates (Stadie et al., 2018; Zheng et al., 2018), and poor generalisation performance (Wu et al., 2018; Yin et al., 2020; Triantafillou et al., 2020).

We argue that defining the meta-learner's objective directly in terms of the learner's objective—i.e. the performance after $K$ update steps—creates two bottlenecks in meta-optimisation. The first bottleneck is curvature: the meta-objective is constrained to the same type of geometry as the learner; the second is myopia: the meta-objective is fundamentally limited to evaluating performance within the $K$-step horizon, but ignores future learning dynamics. Our goal is to design an algorithm that removes these.

The algorithm relies on two main ideas. First, to mitigate myopia, we introduce the notion of bootstrapping a target from the meta-learner itself, a *meta-bootstrap*, that infuses information about learning dynamics in the objective. Second, to control curvature, we formulate the meta-objective in terms of minimising distance (or divergence) to the bootstrapped target, thereby controlling the meta-loss landscape. In this way, the meta-learner learns from its future self. This leads to a bootstrapping effect where improvements beget further improvements. We present a detailed formulation in Section 3; on a high level, as in previous works, we first unroll the meta-learned update rule for $K$ steps to obtain the learner's new parameters. Whereas standard meta-objectives optimise the update rule with respect to (w.r.t.) the learner's performance under the new parameters, our proposed algorithm constructs the meta-objective in two steps:

1. It *bootstraps* a *target* from the learner's new parameters. In this paper, we generate targets by continuing to update the learner's parameters—either under the meta-learned update rule or another update rule—for some number of steps.
2. The learner's new parameters—which are a function of the meta-learner's parameters—and the target are projected onto a *matching space*. A simple example is Euclidean parameter space. To control curvature, we may choose a different (pseudo-)metric space. For instance, a common choice under probabilistic models is the Kullback-Leibler (KL) divergence.

The meta-learner is optimised by minimising distance to the bootstrapped target. We focus on gradient-based optimisation, but other optimisation routines are equally applicable. By optimising meta-parameters in a well-behaved space, we can drastically reduce ill-conditioning and other phenomena that disrupt meta-optimisation. In particular, this form of *Bootstrapped Meta-Gradient* (BMG) enables us to infuse information about *future* learning dynamics without increasing the number of update steps to backpropagate through. In effect, the meta-learner becomes its own teacher. We show that BMG can guarantee performance improvements (Theorem 1) and that this guarantee can be stronger than under standard meta-gradients (Corollary 1). Empirically, we find that BMG provides substantial performance improvements over standard meta-gradients in various settings. We obtain a new state-of-the-art result for model-free agents on Atari (Section 5.2) and improve upon MAML (Finn et al., 2017) in the few-shot setting (Section 6). Finally, we demonstrate how BMG enables new forms of meta-learning, exemplified by meta-learning $\varepsilon$-greedy exploration (Section 5.1).

## 2 RELATED WORK

Bootstrapping as used here stems from temporal difference (TD) algorithms in reinforcement learning (RL) (Sutton, 1988). In these algorithms, an agent learns a value function by using its own future predictions as targets. Bootstrapping has recently been introduced in the self-supervised setting (Guo et al., 2020; Grill et al., 2020). In this paper, we introduce the idea of bootstrapping in the context of meta-learning, where a meta-learner learns about an update rule by generating future targets from it.

Our approach to target matching is related to methods in multi-task meta-learning (Flennerhag et al., 2019; Nichol et al., 2018) that meta-learn an initialisation for SGD by minimising the Euclidean distance to task-optimal parameters. BMG generalise this concept by allowing for arbitrary meta-parameters, matching functions, and target bootstraps. It is further related the more general concept of self-referential meta-learning (Schmidhuber, 1987; 1993), where the meta-learned update rule is used to optimise its own meta-objective.

Target matching under KL divergences results in a form of distillation (Hinton et al., 2015), where an online network (student) is encouraged to match a target network (teacher). In a typical setup, the target is either a fixed (set of) expert(s) (Hinton et al., 2015; Rusu et al., 2015) or a moving aggregation of current experts (Teh et al., 2017; Grill et al., 2020), whereas BMG bootstraps a target by following an update rule. Finally, BMG is loosely inspired by trust-region methods that introduce a distance function to regularize gradient updates (Pascanu & Bengio, 2014; Schulman et al., 2015; Tomar et al., 2020; Hessel et al., 2021).

## 3 BOOTSTRAPPED META-GRADIENTS

We begin in the single-task setting and turn to multi-task meta-learning in Section 6. The learner's problem is to minimize a stochastic objective $f(\mathbf{x}) := \mathbb{E}[\ell(\mathbf{x}; \boldsymbol{\zeta})]$ over a data distribution $p(\boldsymbol{\zeta})$, where $\boldsymbol{\zeta}$ denotes a source of data and $\mathbf{x} \in \mathcal{X} \subset \mathbb{R}^{n_x}$ denotes the learner's parameters. In RL, $f$ is typically the (negative) expected value of a policy $\pi_{\mathbf{x}}$; in supervised learning, $f$ may be the expected negative log-likelihood under a probabilistic model $\pi_{\mathbf{x}}$. We provide precise formulations in Sections 5 and 6.

The meta-learner's problem is to learn an update rule $\varphi : \mathcal{X} \times \mathcal{H} \times \mathcal{W} \to \mathcal{X}$ that updates the learner's parameters by $\mathbf{x}^{(1)} = \mathbf{x} + \varphi(\mathbf{x}, \mathbf{h}, \mathbf{w})$ given $\mathbf{x} \in \mathcal{X}$, a learning state $\mathbf{h} \in \mathcal{H}$, and meta-parameters $\mathbf{w} \in \mathcal{W} \subset \mathbb{R}^{n_w}$ of the update rule. We make no assumptions on the update rule other than differentiability in $\mathbf{w}$. As such, $\varphi$ can be a recurrent neural network (Hochreiter et al., 2001; Wang et al., 2016; Andrychowicz et al., 2016) or gradient descent (Bengio, 2000; Maclaurin et al., 2015; Finn et al., 2017). The learning state $\mathbf{h}$ contains any other data required to compute the update; in a black-box setting $\mathbf{h}$ contains an observation and the recurrent state of the network; for gradient-based updates, $\mathbf{h}$ contains the (estimated) gradient of $f$ at $\mathbf{x}$ along with any auxiliary information; for instance, SGD is given by $\mathbf{x}^{(1)} = \mathbf{x} - \alpha \nabla_x f(\mathbf{x})$ with $\mathbf{h} = \nabla_x f(\mathbf{x})$, $\mathbf{w} = \alpha \in \mathbb{R}_+$.

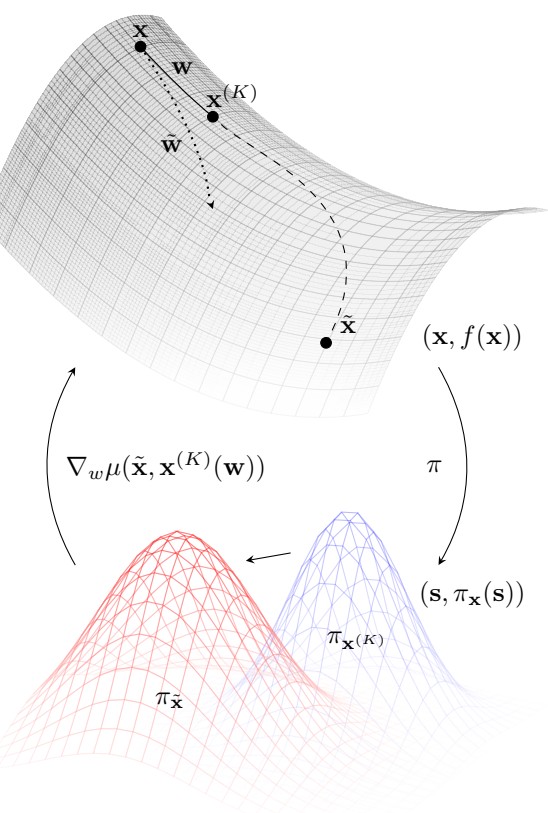

Figure 1: Bootstrapped Meta-Gradients.

The standard meta-gradient (MG) optimises meta-parameters $\mathbf{w}$ by taking $K$ steps under $\varphi$ and evaluating the resulting learner parameter vector under $f$. With a slight abuse of notation, let $\mathbf{x}^{(K)}(\mathbf{w})$ denote the learner's parameters after $K$ applications of $\varphi$ starting from some $(\mathbf{x}, \mathbf{h}, \mathbf{w})$, where $(\mathbf{x}, \mathbf{h})$ evolve according to $\varphi$ and the underlying data distribution. The MG update is defined by

$$\mathbf{w}' = \mathbf{w} - \beta \, \nabla_{\mathbf{w}} f\left(\mathbf{x}^{(K)}(\mathbf{w})\right), \qquad \beta \in \mathbb{R}_+ \,. \tag{1}$$

Extensions involve averaging the performance over all iterates $\mathbf{x}^{(1)}, \ldots, \mathbf{x}^{(K)}$ (Andrychowicz et al., 2016; Chen et al., 2016; Antoniou et al., 2019) or using validation data in the meta-objective (Bengio et al., 1991; Maclaurin et al., 2015; Finn et al., 2017; Xu et al., 2018). We observe two bottlenecks in the meta-objective in Eq. 1. First, the meta-objective is subject to the same curvature as the learner. Thus if $f$ is ill-conditioned, so will the meta-objective be. Second, the meta-objective is only able to evaluate the meta-learner on dynamics up to the $K$th step, but ignores effects of future updates.

To tackle myopia, we introduce a *Target Bootstrap* (TB) $\xi : \mathcal{X} \mapsto \mathcal{X}$ that maps the meta-learner's output $\mathbf{x}^{(K)}$ into a bootstrapped target $\tilde{\mathbf{x}} = \xi(\mathbf{x}^{(K)})$. We focus on TBs that unroll $\varphi$ a further $L - 1$ steps before taking final gradient step on $f$, with targets of the form $\tilde{\mathbf{x}} = \mathbf{x}^{(K+L-1)} - \alpha \nabla f(\mathbf{x}^{(K+L-1)})$. This TB encourages the meta-learner to reach future states on its trajectory faster while nudging the trajectory in a descent direction. Crucially, regardless of the bootstrapping strategy, we do *not* backpropagate through the target. Akin to temporal difference learning in RL (Sutton, 1988), the target is a fixed goal that the meta-learner should try to produce within the $K$-step budget.

Finally, to improve the meta-optimisation landscape, we introduce a *matching function* $\mu : \mathcal{X} \times \mathcal{X} \to \mathbb{R}_+$ that measures the (dis)similarity between the meta-learner's output, $\mathbf{x}^{(K)}(\mathbf{w})$, and the target, $\tilde{\mathbf{x}}$, in a matching space defined by $\mu$ (see Figure 1). Taken together, the BMG update is defined by

$$\tilde{\mathbf{w}} = \mathbf{w} - \beta \, \nabla_{\mathbf{w}} \mu\left(\tilde{\mathbf{x}}, \mathbf{x}^{(K)}(\mathbf{w})\right), \qquad \beta \in \mathbb{R}_+, \tag{2}$$

where the gradient is with respect to the second argument of $\mu$. Thus, BMG describes a family of algorithms based on the choice of matching function $\mu$ and TB $\xi$. In particular, MG is a special case of BMG under matching function $\mu(\tilde{\mathbf{x}}, \mathbf{x}^{(K)}) = \|\tilde{\mathbf{x}} - \mathbf{x}^{(K)}\|_2^2$ and TB $\xi(\mathbf{x}^{(K)}) = \mathbf{x}^{(K)} - \frac{1}{2}\nabla_x f(\mathbf{x}^{(K)})$, since the bootstrapped meta-gradient reduces to the standard meta-gradient:

$$\nabla_w \left\| \tilde{\mathbf{x}} - \mathbf{x}^{(K)}(\mathbf{w}) \right\|_2^2 = -2D\left(\tilde{\mathbf{x}} - \mathbf{x}^{(K)}\right) = D\nabla_x f\left(\mathbf{x}^{(K)}\right) = \nabla_w f\left(\mathbf{x}^{(K)}(\mathbf{w})\right), \qquad (3)$$

where $D$ denotes the (transposed) Jacobian of $\mathbf{x}^{(K)}(\mathbf{w})$. For other matching functions and target strategies, BMG produces different meta-updates compared to MG. We discuss these choices below.

**Matching Function**   Of primary concern to us are models that output a probabilistic distribution, $\pi_\mathbf{x}$. A common pseudo-metric over a space of probability distributions is the Kullback-Leibler (KL) divergence. For instance, Natural Gradients (Amari, 1998) point in the direction of steepest descent under the KL-divergence, often approximated through a KL-regularization term (Pascanu & Bengio, 2014). KL-divergences also arise naturally in RL algorithms (Kakade, 2001; Schulman et al., 2015; 2017; Abdolmaleki et al., 2018). Hence, a natural starting point is to consider KL-divergences between the target and the iterate, e.g. $\mu(\tilde{\mathbf{x}}, \mathbf{x}^{(K)}) = \mathrm{KL}\left(\pi_{\tilde{\mathbf{x}}} \,\|\, \pi_{\mathbf{x}^{(K)}}\right)$. In actor-critic algorithms (Sutton et al., 1999), the policy defines only part of the agent—the value function defines the other. Thus, we also consider a composite matching function over both policy and value function.

**Target Bootstrap**   We analyze conditions under which BMG guarantees performance improvements in Section 4 and find that the target should co-align with the gradient direction. Thus, in this paper we focus on gradient-based TBs and find that they perform well empirically. As with matching functions, this is a small subset of all possible choices; we leave the exploration of other choices for future work.

## 4   PERFORMANCE GUARANTEES

In this analysis, we restrict attention to the noise-less setting (true expectations). In this setting, we ask three questions: (1) what local performance guarantees are provided by MG? (2) What performance guarantees can BMG provide? (3) How do these guarantees relate to each other? To answer these questions, we analyse how the performance around $f(\mathbf{x}^{(K)}(\mathbf{w}))$ changes by updating $\mathbf{w}$ either under standard meta-gradients (Eq. 1) or bootstrapped meta-gradients (Eq. 2).

First, consider improvements under the MG update. In online optimisation, the MG update can achieve strong convergence guarantees if the problem is well-behaved (van Erven & Koolen, 2016), with similar guarantees in the multi-task setting (Balcan et al., 2019; Khodak et al., 2019; Denevi et al., 2019). A central component of these results is that the MG update guarantees a local improvement in the objective. Lemma 1 below presents this result in our setting, with the following notation: let $\|\mathbf{u}\|_A := \sqrt{\langle \mathbf{u}, A\,\mathbf{u}\rangle}$ for any square real matrix $A$. Let $G^T = D^T D \in \mathbb{R}^{n_x \times n_x}$, with $D := \left[\frac{\partial}{\partial \mathbf{w}} \mathbf{x}^{(K)}(\mathbf{w})\right]^T \in \mathbb{R}^{n_w \times n_x}$. Note that $\nabla_w f(\mathbf{x}^{(K)}(\mathbf{w})) = D\nabla_x f(\mathbf{x}^{(K)})$.

**Lemma 1** (MG Descent). *Let $\mathbf{w}'$ be given by Eq. 1. For $\beta$ sufficiently small, $f\left(\mathbf{x}^{(K)}(\mathbf{w}')\right) - f\left(\mathbf{x}^{(K)}(\mathbf{w})\right) = -\beta\|\nabla_x f(\mathbf{x}^{(K)})\|_{G^T}^2 + O(\beta^2) < 0$.*

We defer all proofs to Appendix A. Lemma 1 relates the gains obtained under standard meta-gradients to the local gradient norm of the objective. Because the meta-objective is given by $f$, the MG update is not scale-free (c.f. Schraudolph, 1999), nor invariant to re-parameterisation. If $f$ is highly non-linear, the meta-gradient can vary widely, preventing efficient performance improvement. Next, we turn to BMG, where we assume $\mu$ is differentiable and convex, with $0$ being its minimum.

**Theorem 1** (BMG Descent). *Let $\tilde{\mathbf{w}}$ be given by Eq. 2 for some TB $\xi$. The BMG update satisfies*

$$f\left(\mathbf{x}^{(K)}(\tilde{\mathbf{w}})\right) - f\left(\mathbf{x}^{(K)}(\mathbf{w})\right) = \frac{\beta}{\alpha}\left(\mu(\tilde{\mathbf{x}}, \mathbf{x}^{(K)} - \alpha G^T\,\mathbf{g}) - \mu(\tilde{\mathbf{x}}, \mathbf{x}^{(K)})\right) + o(\beta(\alpha + \beta)).$$

*For $(\alpha, \beta)$ sufficiently small, there exists infinitely many $\xi$ for which $f\left(\mathbf{x}^{(K)}(\tilde{\mathbf{w}})\right) - f\left(\mathbf{x}^{(K)}(\mathbf{w})\right) < 0$. In particular, $\xi(\mathbf{x}^{(K)}) = \mathbf{x}^{(K)} - \alpha G^T\,\mathbf{g}$ yields improvements*

$$f\left(\mathbf{x}^{(K)}(\tilde{\mathbf{w}})\right) - f\left(\mathbf{x}^{(K)}(\mathbf{w})\right) = -\frac{\beta}{\alpha}\mu(\tilde{\mathbf{x}}, \mathbf{x}^{(K)}) + o(\beta(\alpha + \beta)) < 0.$$

*This is not an optimal rate; there exists infinitely many TBs that yield greater improvements.*

Theorem 1 portrays the inherent trade-off in BMG; targets should align with the local direction of steepest descent, but provide as much learning signal as possible. Importantly, this theorem also establishes that $\mu$ directly controls for curvature as improvements are expressed in terms of $\mu$. While the TB $\xi_G^\alpha(\mathbf{x}^{(K)}) := \mathbf{x}^{(K)} - \alpha G^T \mathbf{g}$ yields performance improvements that are proportional to the meta-loss itself, larger improvements are possible by choosing a TB that carries greater learning signal (by increasing $\mu(\tilde{\mathbf{x}}, \mathbf{x}^{(K)})$). To demonstrate that BMG can guarantee larger improvements to the update rule than MG, we consider the TB $\xi_G^\alpha$ with $\mu$ the (squared) Euclidean norm. Let $r := \|\nabla f(\mathbf{x}^{(K)})\|_2 / \|G^T \nabla f(\mathbf{x}^{(K)})\|_2$ denote the gradient norm ratio.

**Corollary 1.** *Let* $\mu = \|\cdot\|_2^2$ *and* $\tilde{\mathbf{x}} = \xi_G^r(\mathbf{x}^{(K)})$. *Let* $\mathbf{w}'$ *be given by Eq. 1 and* $\tilde{\mathbf{w}}$ *be given by Eq. 2. For* $\beta$ *sufficiently small,* $f(\mathbf{x}^{(K)}(\tilde{\mathbf{w}})) \le f(\mathbf{x}^{(K)}(\mathbf{w}'))$, *strictly if* $GG^T \ne G^T$ *and* $G^T \nabla_x f(\mathbf{x}^{(K)}) = \mathbf{0}$.

**Discussion** Our analysis focuses on an arbitrary (albeit noiseless) objective $f$ and establishes that BMG can guarantee improved performance under a variety of TBs. We further show that BMG can yield larger local improvements than MG. To identify optimal TBs, further assumptions are required on $f$ and $\mu$, but given these Theorem 1 can serve as a starting point for more specialised analysis. Empirically, we find that taking $L$ steps on the meta-learned update with an final gradient step on the objective performs well. Theorem 1 exposes a trade-off for targets that are "far" away. Empirically, we observe clear benefits from bootstraps that unroll the meta-learner for several steps before taking a gradient step on $f$; exploring other forms of bootstraps is an exciting area for future research.

## 5 REINFORCEMENT LEARNING

We consider a typical reinforcement learning problem, modelled as an MDP $\mathcal{M} = (\mathcal{S}, \mathcal{A}, \mathcal{P}, \mathcal{R}, \gamma)$. Given an initial state $\mathbf{s}_0 \in \mathcal{S}$, at each time step $t \in \mathbb{N}$, the agent takes an action $\mathbf{a}_t \sim \pi_{\mathbf{x}}(\mathbf{a} \mid \mathbf{s}_t)$ from a *policy* $\pi : \mathcal{S} \times \mathcal{A} \to [0,1]$ parameterised by $\mathbf{x}$. The agent obtains a reward $r_{t+1} \sim \mathcal{R}(\mathbf{s}_t, \mathbf{a}_t, \mathbf{s}_{t+1})$ based on the transition $\mathbf{s}_{t+1} \sim \mathcal{P}(\mathbf{s}_{t+1} \mid \mathbf{s}_t, \mathbf{a}_t)$. The *action-value* of the agent's policy given a state $\mathbf{s}_0$ and action $\mathbf{a}_0$ is given by $Q_{\mathbf{x}}(\mathbf{s}_0, \mathbf{a}_0) := \mathbb{E}[\sum_{t=0}^{\infty} \gamma^t r_{t+1} \mid \mathbf{s}_0, \mathbf{a}_0, \pi_{\mathbf{x}}]$ under discount rate $\gamma \in [0,1)$. The corresponding *value* of policy $\pi_{\mathbf{x}}$ is given by $V_{\mathbf{x}}(\mathbf{s}_0) := \mathbb{E}_{\mathbf{a}_0 \sim \pi_{\mathbf{x}}(\mathbf{a} \mid \mathbf{s}_0)}[Q_{\mathbf{x}}(\mathbf{s}_0, \mathbf{a}_0)]$.

The agent's problem is to learn a policy that maximises the value given an expectation over $\mathbf{s}_0$, defined either by an initial state distribution in the episodic setting (e.g. Atari, Section 5.2) or the stationary state-visitation distribution under the policy in the non-episodic setting (Section 5.1). Central to RL is the notion of *policy-improvement*, which takes a current policy $\pi_{\mathbf{x}}$ and constructs a new policy $\pi_{\mathbf{x}'}$ such that $\mathbb{E}[V_{\mathbf{x}'}] \ge \mathbb{E}[V_{\mathbf{x}}]$. A common policy-improvement step is $\arg\max_{\mathbf{x}'} \mathbb{E}_{a \sim \pi_{\mathbf{x}'}(a|s)}[Q_{\mathbf{x}}(s, a)]$.

Most works in meta-RL rely on actor-critic algorithms (Sutton et al., 1999). These treat the above policy-improvement step as an optimisation problem and estimate a *policy-gradient* (Williams & Peng, 1991; Sutton et al., 1999) to optimise $\mathbf{x}$. To estimate $V_{\mathbf{x}}$, these introduce a *critic* $v_{\mathbf{z}}$ that is jointly trained with the policy. The policy is optimised under the current estimate of its value function, while the critic is tracking the value function by minimizing a Temporal-Difference (TD) error. Given a *rollout* $\tau = (\mathbf{s}_0, \mathbf{a}_0, r_1, \mathbf{s}_1, \dots, r_T, \mathbf{s}_T)$, the objective is given by $f(\mathbf{x}, \mathbf{z}) = \epsilon_{\text{PG}} \ell_{\text{PG}}(\mathbf{x}) + \epsilon_{\text{EN}} \ell_{\text{EN}}(\mathbf{x}) + \epsilon_{\text{TD}} \ell_{\text{TD}}(\mathbf{z}), \epsilon_{\text{PG}}, \epsilon_{\text{EN}}, \epsilon_{\text{TD}} \in \mathbb{R}_+$, where

$$\ell_{\text{EN}}(\mathbf{x}) = \sum_{t \in \tau} \sum_{\mathbf{a} \in \mathcal{A}} \pi_{\mathbf{x}}(\mathbf{a} \mid \mathbf{s}_t) \log \pi_{\mathbf{x}}(\mathbf{a} \mid \mathbf{s}_t), \qquad \ell_{\text{TD}}(\mathbf{z}) = \frac{1}{2} \sum_{t \in \tau} \left( G_t^{(n)} - v_{\mathbf{z}}(\mathbf{s}_t) \right)^2,$$

$$\ell_{\text{PG}}(\mathbf{x}) = -\sum_{t \in \tau} \rho_t \log \pi_{\mathbf{x}}(\mathbf{a}_t \mid \mathbf{s}_t) \left( G_t^{(n)} - v_{\mathbf{z}}(\mathbf{s}_t) \right),$$

(4)

where $\rho_t$ denotes an importance weight and $G_t^{(n)}$ denotes an $n$-step bootstrap target. Its form depends on the algorithm; in Section 5.1, we generate rollouts from $\pi_{\mathbf{x}}$ (on-policy), in which case $\rho_t = 1$ and $G_t^{(n)} = \sum_{i=0}^{(n-1)} \gamma^i r_{t+i+1} + \gamma^n v_{\bar{\mathbf{z}}}(\mathbf{s}_{t+n}) \forall t$, where $\bar{\mathbf{z}}$ denotes fixed (non-differentiable) parameters. In the off-policy setting (Section 5.2), $\rho$ corrects for sampling bias and $G_t^{(n)}$ is similarly adjusted.

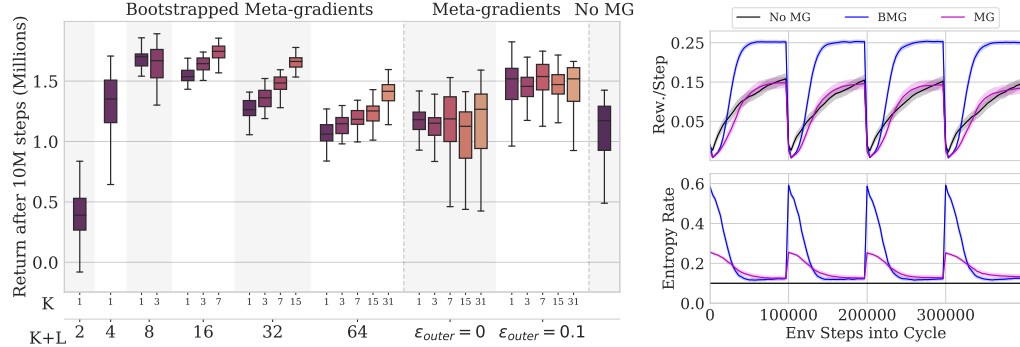

Figure 2: Non-stationary grid-world (Section 5.1). Left: Comparison of total returns under an actor-critic agent over 50 seeds. Right: Learned entropy-regularization schedules. The figure depicts the average regularization weight ($\epsilon$) over 4 task-cycles at 6M steps in the environment.

## 5.1 A NON-STATIONARY AND NON-EPISODIC GRID WORLD

We begin with a tabular grid-world with two items to collect. Once an item is collected, it is randomly re-spawned. One item yields a reward of $+1$ and the other a reward of $-1$. The reward is flipped every 100,000 steps. To succeed, a memory-less agent must efficiently re-explore the environment. We study an on-policy actor-critic agent with $\epsilon_{PG} = \epsilon_{TD} = 1$. As baseline, we tune a fixed entropy-rate weight $\epsilon = \epsilon_{EN}$. We compare against agents that meta-learn $\epsilon$ online. For MG, we use the actor-critic loss as meta-objective ($\epsilon$ fixed), as per Eq. 1. The setup is described in full in Appendix B.1

**BMG** Our primary focus is on the effect of bootstrapping. Because this setup is fully online, we can generate targets using the most recent $L-1$ parameter updates and a final agent parameter update using $\epsilon = 0$. Hence, the computational complexity

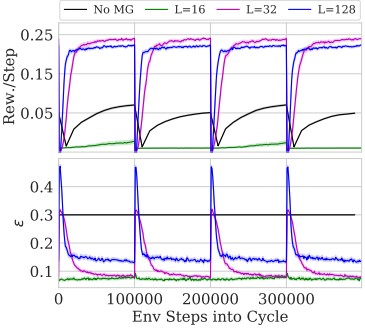

Figure 3: BMG $\varepsilon$-greedy exploration under a $Q(\lambda)$-agent.

of BMG is constant in $L$ under this implementation (see Appendix B.2). We define the matching function as the KL-divergence between $\mathbf{x}^{(K)}$ and the target, $\mu(\tilde{\mathbf{x}}, \mathbf{x}^{(K)}(\mathbf{w})) = \mathrm{KL}\left(\pi_{\tilde{\mathbf{x}}} \| \pi_{\mathbf{x}^{(K)}}\right)$.

Figure 2 presents our main findings. Both MG and BMG learn adaptive entropy-rate schedules that outperform the baseline. However, MG fails if $\epsilon = 0$ in the meta-objective, as it becomes overly greedy (Figure 9). MG shows no clear benefit of longer meta-learning horizons, indicating that myopia stems from the objective itself. In contrast, BMG exhibits greater adaptive capacity and is able to utilise greater meta-learning horizons. Too short horizons induce myopia, whereas too long prevent efficient adaptation. For a given horizon, increasing $K$ is uniformly beneficial. Finally, we find that BMG outperforms MG for a given horizon without backpropagating through all updates. For instance, for $K = 8$, BMG outperforms MG with $K = 1$ and $L = 7$. Our ablation studies (Appendix B.2) show that increasing the target bootstrap length counters myopia; however, using the meta-learned update rule for all $L$ steps can derail meta-optimization.

Next, we consider a new form of meta-learning: learning $\varepsilon$-greedy exploration in a $Q(\lambda)$-agent (precise formulation in Appendix B.3). While the $\varepsilon$ parameter has a similar effect to entropy-regularization, $\varepsilon$ is a parameter applied in the behaviour-policy while acting. As it does not feature in the loss function, it is not readily optimized by existing meta-gradient approaches. In contrast, BMG can be implemented by matching the policy derived from a target action-value function, precisely as in the actor-critic case. An implication is that BMG can meta-learn *without* backpropagating through the update rule. Significantly, this opens up to meta-learning (parts of) the *behaviour policy*, which is hard to achieve in the MG setup as the behaviour policy is not used in the update rule. Figure 3 shows that meta-learning $\varepsilon$-greedy exploration in this environment significantly outperforms the best fixed $\varepsilon$ found by hyper-parameter tuning. As in the actor-critic case, we find that BMG responds positively to longer meta-learning horizons (larger $L$); see Appendix B.3, Figure 12 for detailed results.

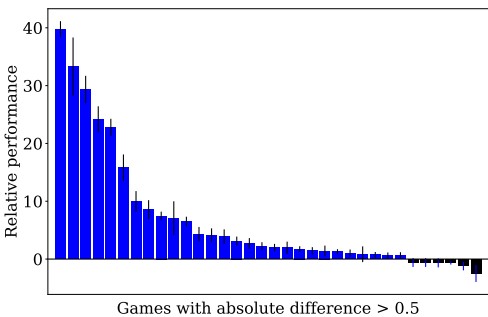 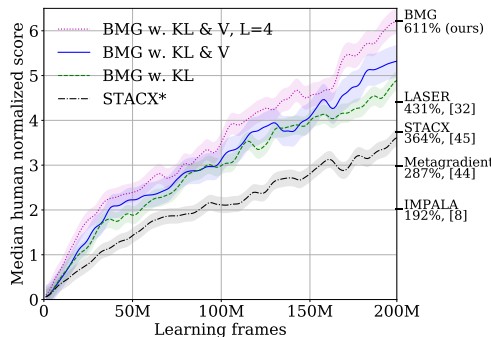

Figure 4: Human-normalized score across the 57 games in Atari ALE. Left: per-game difference in score between BMG and our implementation of STACX* at 200M frames. Right: Median scores over learning compared to published baselines. Shading depict standard deviation across 3 seeds.

## 5.2 ATARI

High-performing RL agents tend to rely on distributed learning systems to improve data efficiency (Kapturowski et al., 2018; Espeholt et al., 2018). This presents serious challenges for meta-learning as the policy gradient becomes noisy and volatile due to off-policy estimation (Xu et al., 2018; Zahavy et al., 2020). Theorem 1 suggests that BMG can be particularly effective in this setting under the appropriate distance function. To test these predictions, we adapt the Self-Tuning Actor-Critic (STACX; Zahavy et al., 2020) to meta-learn under BMG on the 57 environments in the Atari Arcade Learning Environment (ALE; Bellemare et al., 2013).

**Protocol** We follow the original IMPALA setup (Espeholt et al., 2018), but we do not downsample or gray-scale inputs. Following the literature, we train for 200 million frames and evaluate agent performance by median Human Normalized Score (HNS) across 3 seeds (Espeholt et al., 2018; Xu et al., 2018; Zahavy et al., 2020).

**STACX** The IMPALA actor-critic agent runs multiple actors asynchronously to generate experience for a centralized learner. The learner uses truncated importance sampling to correct for off-policy data in the actor-critic update, which adjusts $\rho$ and $\hat{V}$ in Eq. 4. The STACX agent (Zahavy et al., 2020) is a state-of-the-art meta-RL agent. It builds on IMPALA in two ways: (1) it introduces auxiliary tasks in the form of additional objectives that differ only in their hyper-parameters; (2) it meta-learns the hyper-parameters of each loss function (main and auxiliary). Meta-parameters are given by $\mathbf{w} = (\gamma^i, \epsilon_{\text{PG}}^i, \epsilon_{\text{EN}}^i, \epsilon_{\text{TD}}^i, \lambda^i, \alpha^i)_{i=1}^{1+n}$, where $\lambda$ and $\alpha$ are hyper-parameters of the importance weighting mechanism and $n = 2$ denotes the number of auxiliary tasks. STACX uses the IMPALA objective as the meta-objective with $K = 1$. See Appendix C for a complete description.

**BMG** We conduct ceteris-paribus comparisons that only alter the meta-objective: agent parameter updates are *identical* to those in STACX. When $L = 1$, the target takes a gradient step on the original IMPALA loss, and hence the *only* difference is the form of the meta-objective; they both use the same data and gradient information. For $L > 1$, the first $L - 1$ steps bootstrap from the meta-learned update rule itself. To avoid overfitting, each of the $L - 1$ steps use separate replay data; this extra data is not used anywhere else. To understand matching functions, we test policy matching and value matching. Policy matching is defined by $\mu(\tilde{\mathbf{x}}, \mathbf{x}^{(K)}(\mathbf{w})) = \text{KL}\left(\pi_{\tilde{\mathbf{x}}} \| \pi_{\mathbf{x}^{(1)}}\right)$; we also test a symmetric KL-divergence (KL-S). Value matching is defined by $\mu(\tilde{\mathbf{z}}, \mathbf{z}^{(1)}(\mathbf{w})) := \mathbb{E}\left[(v_{\tilde{z}} - v_{z^{(1)}})^2\right]$.

Figure 4 presents our main comparison. BMG with $L = 1$ and policy-matching (KL) obtains a median HNS of ~500%, compared to ~350% for STACX. Recall that for $L = 1$, BMG uses the same data to compute agent parameter update, target update, and matching loss; hence this is an apples-to-apples comparison. Using both policy matching and value matching (with 0.25 weight on the latter) further improves the score to ~520% and outperforms STACX across almost all 57 games, with a few minor exceptions (left panel, Figure 4). These results are obtained *without* tuning hyper-parameters for BMG. Finally, extending the meta-learning horizon by setting $L = 4$ and adjusting gradient clipping from .3 to .2 obtains a score of ~610%.

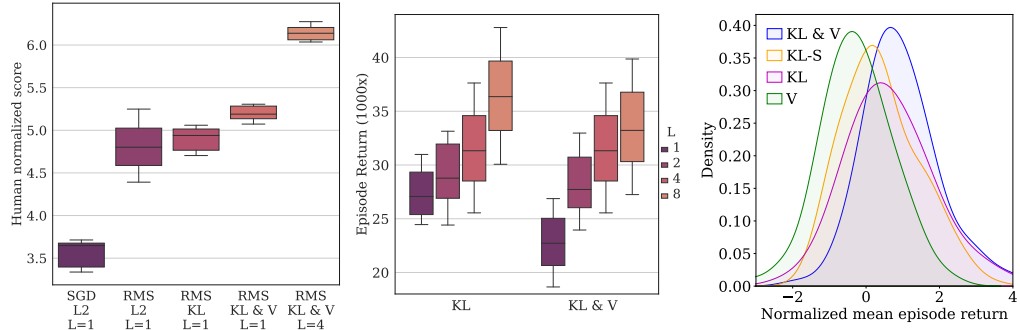

Figure 5: Ablations on Atari. Left: human normalized score decomposition of TB w.r.t. optimizer (SGD, RMS), matching function (L2, KL, KL & V), and bootstrap steps ($L$). BMG with (SGD, L2, $L=1$) is equivalent to STACX. Center: episode return on Ms Pacman for different $L$. Right: distribution of episode returns over all 57 games, normalized per-game by mean and standard deviation. All results are reported between 190-200M frames over 3 independent seeds.

In Figure 5, we turn to ablations. In the left-panel, we deconstruct BMG into STACX (i.e., MG) and compare performances. We find that roughly 45% of the performance gains comes from curvature correction (given by using RMSProp in the target bootstrap). The matching function can further control curvature to obtain performance improvements, accounting for roughly 25%. Finally, increasing $L$, thereby reducing myopia, accounts for about 30% of the performance improvement. Comparing the cosine similarity between consecutive meta-gradients, we find that BMG improves upon STACX by two orders of magnitude. Detailed ablations in Appendix C.1.

The center panel of Figure 5 provides a deep-dive in the effect of increasing the meta-learning horizon ($L > 1$) in Ms Pacman. Performance is uniformly increasing in $L$, providing further support that BMG can increase the effective meta-horizon without increasing the number of update steps to backpropagate through. A more in-depth analysis Appendix C.3 reveals that $K$ is more sensitive to curvature and the quality of data. However, bootstrapping *only* from the meta-learner for all $L$ steps can lead to degeneracy (Appendix C.2, Figure 14). In terms of replay (Appendix C.2), while standard MG degrades with more replay, BMG benefits from more replay in the target bootstrap.

The right panel of Figure 5 studies the effect of the matching function. Overall, joint policy and value matching exhibits best performance. In contrast to recent work (Tomar et al., 2020; Hessel et al., 2021), we do not find that reversing the KL-direction is beneficial. Using only value-matching results in worse performance, as it does not optimise for efficient policy improvements. Finally, we conduct detailed analysis of scalability in Appendix C.4. While BMG is 20% slower for $K = 1, L = 1$ due to the target bootstrap, it is 200% faster when MG uses $K = 4$ and BMG uses $K = 1, L = 3$.

## 6 MULTI-TASK FEW-SHOT LEARNING

Multi-task meta-learning introduces an expectation over task objectives. BMG is applied by computing task-specific bootstrap targets, with the meta-gradient being the expectation over task-specific matching losses. For a general multi-task formulation, see Appendix D; here we focus on the few-shot classification paradigm. Let $f_{\mathcal{D}} : \mathcal{X} \to \mathbb{R}$ denote the negative log-likelihood loss on some data $\mathcal{D}$. A *task* is defined as a pair of datasets $(\mathcal{D}_{\tau}, \mathcal{D}'_{\tau})$, where $\mathcal{D}_{\tau}$ is a training set and $\mathcal{D}'_{\tau}$ is a validation set. In the $M$-shot-$N$-way setting, each task has $N$ classes and $\mathcal{D}_{\tau}$ contains $M$ observations per class.

The goal of this experiment is to study how the BMG objective behaves in the multi-task setting. For this purpose, we focus on the canonical MAML setup (Finn et al., 2017), which meta-learns an initialisation $\mathbf{x}_{\tau}^{(0)} = \mathbf{w}$ for SGD that is shared across a task distribution $p(\tau)$. Adaptation is defined by $\mathbf{x}_{\tau}^{(k)} = \mathbf{x}_{\tau}^{(k-1)} + \alpha \nabla f_{\mathcal{D}_{\tau}}(\mathbf{x}_{\tau}^{(k-1)})$, with $\alpha \in \mathbb{R}_{+}$ fixed. The meta-objective is the validation loss in expectation over the task distribution: $\mathbb{E}[f_{\mathcal{D}'_{\tau}}(\mathbf{x}_{\tau}^{(K)}(\mathbf{w}))]$. Several works have extended this setup by altering the update rule ($\varphi$) (Lee & Choi, 2018; Zintgraf et al., 2019; Park & Oliva, 2019; Flennerhag et al., 2020). As our focus is on the meta-objective, we focus on comparisons with MAML.

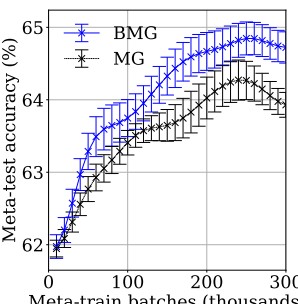 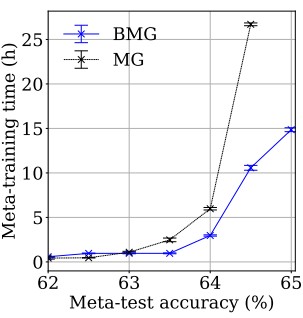 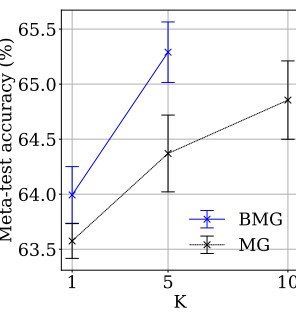

Figure 6: MiniImagenet 5-way-5-shot meta-test performance. Left: performance as a function of meta-training batches. Center: performance as a function of wall-clock time. Right: best reported performance under each $K$. Error bars depict standard deviation across 3 seeds.

**BMG** For each task, a target $\tilde{\mathbf{x}}_\tau$ is bootstrapped by taking $L$ SGD steps from $\mathbf{x}_\tau^{(K)}$ using validation data. The BMG objective is the expected distance, $\mathbb{E}[\mu(\tilde{\mathbf{x}}_\tau, \mathbf{x}_\tau^{(K)})]$. The KL-divergence as matching function has an interesting connection to MG. The target $\tilde{\mathbf{x}}_\tau$ can be seen as an "expert" on task $\tau$ so that BMG is a form of distillation (Hinton et al., 2015). The log-likelihood loss used by MG is also a KL divergence, but w.r.t. a "cold" expert that places all mass on the true label. Raising the temperature in the target can allow BMG to transfer more information (Hinton & Plaut, 1987).

**Setup** We use the MiniImagenet benchmark (Vinyals et al., 2016) and study two forms of efficiency: for *data efficiency*, we compare meta-test performance as function of the number of meta-training batches; for *computational efficiency*, we compare meta-test performance as a function of training time. To reflect what each method would achieve for a given computational budget, we report meta-test performance for the hyper-parameter configuration with best meta-validation performance. For MG, we tune the meta-learning rate $\beta \in \{10^{-3}, 10^{-4}\}$, $K \in \{1, 5, 10\}$, and options to use first-order approximations ((FOMAML; Finn et al., 2017) or (ANIL; Raghu et al., 2020)). For BMG, we tune $\beta \in \{10^{-3}, 10^{-4}\}$, $K \in \{1, 5\}$, as well as $L \in \{1, 5, 10\}$, and the direction of the KL.

The left panel of Figure 6 presents results on data efficiency. For few meta-updates, MG and BMG are on par. For 50 000 meta-updates and beyond, BMG achieves strictly superior performance, with the performance delta increasing over meta-updates. The central panel presents results on computational efficiency; we plot the time required to reach a given meta-test performance. This describes the relationship between performance and computational complexity. We find BMG exhibits better scaling properties, reaching the best performance of MG in approximately half the time. Finally, in the right panel, we study the effect of varying $K$. BMG achieves higher performance for both $K = 1$ and $K = 5$. We allow MG to also use $K = 10$, but this did not yield any significant gains. We conduct an analysis of the impact BMG has on curvature and meta-gradient variance in Appendix D.3. To summarise, we find that BMG significantly improves upon the MG meta-objective, both in terms of data efficiency, computational efficiency, and final performance.

## 7 CONCLUSION

In this paper, we have put forth the notion that efficient meta-learning does not require the meta-objective to be expressed directly in terms of the learner's objective. Instead, we present an alternative approach that relies on having the meta-learner match a desired target. Here, we bootstrap from the meta-learned update rule itself to produce future targets. While using the meta-learned update rule as the bootstrap allows for an open-ended meta-learning process, some grounding is necessary. As an instance of this approach, we study bootstrapped meta-gradients, which can guarantee performance improvements under appropriate choices of targets and matching functions that can be larger than those of standard meta-gradients. Empirically, we observe substantial improvements on Atari and achieve a new state-of-the-art, while obtaining significant efficiency gains in a multi-task meta-learning setting. We explore new possibilities afforded by the target-matching nature of the algorithm and demonstrate that it can learn to explore in an $\epsilon$-greedy $Q$-learning agent.

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

# Bootstrapped Meta-Learning: Appendix

## CONTENTS

## A  PROOFS

This section provides complete proofs for the results in Section 4. Throughout, we assume that $(\mathbf{x}^{(0)}, \mathbf{h}^{(0)}, \mathbf{w})$ is given and write $\mathbf{x} := \mathbf{x}^{(0)}$, $\mathbf{h} := \mathbf{h}^{(0)}$. We assume that $\mathbf{h}$ evolves according to some process that maps a history $H^{(k)} := (\mathbf{x}^{(0)}, \mathbf{h}^{(0)}, \ldots, \mathbf{x}^{(k-1)}, \mathbf{h}^{(k-1)}, \mathbf{x}^{(k)})$ into a new learner state $\mathbf{h}^{(k)}$, including any sampling of data (c.f. Section 3). Recall that we restrict attention to the noiseless setting, and hence updates are considered in expectation. We define the *map* $\mathbf{x}^{(K)}(\mathbf{w})$ by

$$\mathbf{x}^{(1)} = \mathbf{x}^{(0)} + \varphi\big(\mathbf{x}^{(0)}, \mathbf{h}^{(0)}, \mathbf{w}\big)$$
$$\mathbf{x}^{(2)} = \mathbf{x}^{(1)} + \varphi\big(\mathbf{x}^{(1)}, \mathbf{h}^{(1)}, \mathbf{w}\big)$$
$$\vdots$$
$$\mathbf{x}^{(K)} = \mathbf{x}^{(K-1)} + \varphi\big(\mathbf{x}^{(K-1)}, \mathbf{h}^{(K-1)}, \mathbf{w}\big).$$

The derivative $\frac{\partial}{\partial \mathbf{w}} \mathbf{x}^{(K)}(\mathbf{w})$ differentiates through each step of this process (Hochreiter et al., 2001). As previously stated, we assume $f$ is Lipschitz and that $\mathbf{x}^{(K)}$ is Lipschitz w.r.t. $\mathbf{w}$. We are now in a position to prove results from the main text. We re-state them for convenience.

**Lemma 1** (MG Descent). *Let $\mathbf{w}'$ be given by Eq. 1. For $\beta$ sufficiently small, $f\big(\mathbf{x}^{(K)}(\mathbf{w}')\big) - f\big(\mathbf{x}^{(K)}(\mathbf{w})\big) = -\beta\|\nabla_x f(\mathbf{x}^{(K)})\|_{G^T}^2 + o(\beta^2) < 0$.*

*Proof.* Define $\mathbf{g} := \nabla_x f(\mathbf{x}^{(K)}(\mathbf{w}))$. The meta-gradient at $(\mathbf{x}, \mathbf{h}, \mathbf{w})$ is given by $\nabla_w f(\mathbf{x}^{(K)}(\mathbf{w})) = D\,\mathbf{g}$. Under Eq. 1, we find $\mathbf{w}' = \mathbf{w} - \beta D\,\mathbf{g}$. By first-order Taylor Series Expansion of $f$ around $(\mathbf{x}, \mathbf{h}, \mathbf{w}')$ with respect to $\mathbf{w}$:

$$f\big(\mathbf{x}^{(K)}(\mathbf{w}')\big) = f\big(\mathbf{x}^{(K)}(\mathbf{w})\big) + \langle D\,\mathbf{g}, \mathbf{w}' - \mathbf{w}\rangle + o(\beta^2\|\mathbf{g}\|_{G^T}^2)$$
$$= f\big(\mathbf{x}^{(K)}(\mathbf{w})\big) - \beta\langle D\,\mathbf{g}, D\,\mathbf{g}\rangle + o(\beta^2\|\mathbf{g}\|_{G^T}^2)$$
$$= f\big(\mathbf{x}^{(K)}(\mathbf{w})\big) - \beta\|\mathbf{g}\|_{G^T}^2 + o(\beta^2\|\mathbf{g}\|_{G^T}^2),$$

with $\|\mathbf{g}\|_{G^T}^2 \geq 0$ by virtue of positive semi-definiteness of $G$. Hence, for $\beta^2$ small the residual vanishes and the conclusion follows. ∎

**Theorem 1** (BMG Descent). *Let $\tilde{\mathbf{w}}$ be given by Eq. 2 for some TB $\xi$. The BMG update satisfies*

$$f\big(\mathbf{x}^{(K)}(\tilde{\mathbf{w}})\big) - f\big(\mathbf{x}^{(K)}(\mathbf{w})\big) = \frac{\beta}{\alpha}\left(\mu(\tilde{\mathbf{x}}, \mathbf{x}^{(K)} - \alpha G^T\,\mathbf{g}) - \mu(\tilde{\mathbf{x}}, \mathbf{x}^{(K)})\right) + o(\beta(\alpha + \beta)).$$

*For $(\alpha, \beta)$ sufficiently small, there exists infinitely many $\xi$ for which $f\big(\mathbf{x}^{(K)}(\tilde{\mathbf{w}})\big) - f\big(\mathbf{x}^{(K)}(\mathbf{w})\big) < 0$. In particular, $\xi(\mathbf{x}^{(K)}) = \mathbf{x}^{(K)} - \alpha G^T\,\mathbf{g}$ yields improvements*

$$f\big(\mathbf{x}^{(K)}(\tilde{\mathbf{w}})\big) - f\big(\mathbf{x}^{(K)}(\mathbf{w})\big) = -\frac{\beta}{\alpha}\mu(\tilde{\mathbf{x}}, \mathbf{x}^{(K)}) + o(\beta(\alpha + \beta)) < 0.$$

*This is not an optimal rate; there exists infinitely many TBs that yield greater improvements.*

*Proof.* The bootstrapped meta-gradient at $(\mathbf{x}, \mathbf{h}, \mathbf{w})$ is given by

$$\nabla_w \mu\Big(\tilde{\mathbf{x}}, \mathbf{x}^{(K)}(\mathbf{w})\Big) = D\,\mathbf{u}, \quad \text{where} \quad \mathbf{u} := \nabla_z \mu(\tilde{\mathbf{x}}, \mathbf{z})\Big|_{\mathbf{z}=\mathbf{x}^{(K)}}.$$

Under Eq. 2, we find $\tilde{\mathbf{w}} = \mathbf{w} - \beta D\,\mathbf{u}$. Define $\mathbf{g} := \nabla_x f(\mathbf{x}^{(K)})$. By first-order Taylor Series Expansion of $f$ around $(\mathbf{x}, \mathbf{h}, \tilde{\mathbf{w}})$ with respect to $\mathbf{w}$:

$$\begin{aligned}
f\big(\mathbf{x}^{(K)}(\tilde{\mathbf{w}})\big) &= f\big(\mathbf{x}^{(K)}(\mathbf{w})\big) + \langle D\,\mathbf{g}, \tilde{\mathbf{w}} - \mathbf{w}\rangle + o(\beta^2 \|D\,\mathbf{u}\|_2^2) \\
&= f\big(\mathbf{x}^{(K)}(\mathbf{w})\big) - \beta\langle D\,\mathbf{g}, D\,\mathbf{u}\rangle + o(\beta^2 \|D\,\mathbf{u}\|_2^2) \\
&= f\big(\mathbf{x}^{(K)}(\mathbf{w})\big) - \beta\langle \mathbf{u}, G^T\,\mathbf{g}\rangle + o(\beta^2 \|\mathbf{u}\|_{G^T}^2).
\end{aligned} \tag{5}$$

To bound the inner product, expand $\mu(\tilde{\mathbf{x}}, \cdot)$ around a point $\mathbf{x}^{(K)} + \mathbf{d}$, where $\mathbf{d} \in \mathbb{R}^{n_x}$, w.r.t. $\mathbf{x}^{(K)}$:

$$\mu(\tilde{\mathbf{x}}, \mathbf{x}^{(K)} + \mathbf{d}) = \mu(\tilde{\mathbf{x}}, \mathbf{x}^{(K)}) + \langle \mathbf{u}, \mathbf{d}\rangle + o(\|\mathbf{d}\|_2^2).$$

Thus, choose $\mathbf{d} = -\alpha G^T\,\mathbf{g}$, for some $\alpha \in \mathbb{R}_+$ and rearrange to get

$$-\beta\langle \mathbf{u}, G^T\,\mathbf{g}\rangle = \frac{\beta}{\alpha}\Big(\mu(\tilde{\mathbf{x}}, \mathbf{x}^{(K)} - \alpha G^T\,\mathbf{g}) - \mu(\tilde{\mathbf{x}}, \mathbf{x}^{(K)})\Big) + o(\alpha\beta \|\mathbf{g}\|_{G^T}^2).$$

Substitute into Eq. 5 to obtain

$$\begin{aligned}
f\big(\mathbf{x}^{(K)}(\tilde{\mathbf{w}})\big) - f\big(\mathbf{x}^{(K)}(\mathbf{w})\big) &= \frac{\beta}{\alpha}\Big(\mu(\tilde{\mathbf{x}}, \mathbf{x}^{(K)} - \alpha G^T\,\mathbf{g}) - \mu(\tilde{\mathbf{x}}, \mathbf{x}^{(K)})\Big) \\
&\quad + o(\alpha\beta \|\mathbf{g}\|_{G^T}^2 + \beta^2 \|\mathbf{u}\|_{G^T}^2).
\end{aligned} \tag{6}$$

Thus, the BMG update comes out as the difference between to distances. The first distance is a distortion terms that measures how well the target aligns to the tangent vector $-G^T\,\mathbf{g}$, which is the direction of steepest descent in the immediate vicinity of $\mathbf{x}^{(K)}$ (c.f. Lemma 1). The second term measures learning; greater distance carry more signal for meta-learning. The two combined captures the inherent trade-off in BMG; moving the target further away increases distortions from curvature, but may also increase the learning signal. Finally, the residual captures distortions due to curvature.

*Existence.* To show that there always exists a target that guarantees a descent direction, choose $\tilde{\mathbf{x}} = \mathbf{x}^{(K)} - \alpha G^T\,\mathbf{g}$. This eliminates the first distance in Eq. 6 as the target is perfectly aligned the direction of steepest descent and we obtain

$$f\big(\mathbf{x}^{(K)}(\tilde{\mathbf{w}})\big) - f\big(\mathbf{x}^{(K)}(\mathbf{w})\big) = -\frac{\beta}{\alpha}\mu(\tilde{\mathbf{x}}, \mathbf{x}^{(K)}) + o(\beta(\alpha + \beta)).$$

The residual vanishes exponentially fast as $\alpha$ and $\beta$ go to 0. Hence, there is some $(\bar{\alpha}, \bar{\beta}) \in \mathbb{R}_+^2$ such that for any $(\alpha, \beta) \in (0, \bar{\alpha}) \times (0, \bar{\beta})$, $f\big(\mathbf{x}^{(K)}(\tilde{\mathbf{w}})\big) - f\big(\mathbf{x}^{(K)}(\mathbf{w})\big) < 0$. For any such choice of $(\alpha, \beta)$, by virtue of differentiability in $\mu$ there exists some neighborhood $N$ around $\mathbf{x}^{(K)} - \alpha G^T\,\mathbf{g}$ for which any $\tilde{\mathbf{x}} \in N$ satisfy $f\big(\mathbf{x}^{(K)}(\tilde{\mathbf{w}})\big) - f\big(\mathbf{x}^{(K)}(\mathbf{w})\big) < 0$.

*Efficiency.* We are to show that, given $(\alpha, \beta)$, the set of optimal targets does not include $\tilde{\mathbf{x}} = \mathbf{x}^{(K)} - \alpha G^T\,\mathbf{g}$. To show this, it is sufficient to demonstrate that show that this is not a local minimum of the right hand-side in Eq. 6. Indeed,

$$\begin{aligned}
&\nabla_{\tilde{x}}\left(\frac{\beta}{\alpha}\Big(\mu(\tilde{\mathbf{x}}, \mathbf{x}^{(K)} - \alpha G^T\,\mathbf{g}) - \mu(\tilde{\mathbf{x}}, \mathbf{x}^{(K)})\Big) + o(\alpha\beta \|\mathbf{g}\|_{G^T}^2 + \beta^2 \|\mathbf{u}\|_{G^T}^2)\right)\Bigg|_{\tilde{\mathbf{x}}=\mathbf{x}^{(K)} - \alpha G^T\,\mathbf{g}} \\
&= -\frac{\beta}{\alpha}\nabla_{\tilde{x}}\,\mu(\tilde{\mathbf{x}}, \mathbf{x}^{(K)})\Big|_{\tilde{\mathbf{x}}=\mathbf{x}^{(K)} - \alpha G^T\,\mathbf{g}} + \beta^2\,\mathbf{o} \neq \mathbf{0},
\end{aligned}$$

where $\beta^2\,\mathbf{o}$ is the gradient of the residual ($\|\mathbf{u}\|_2^2$ depends on $\tilde{\mathbf{x}}$) w.r.t. $\tilde{\mathbf{x}} = \mathbf{x}^{(K)} - \alpha G^T\,\mathbf{g}$. To complete the proof, let $\tilde{\mathbf{u}}$ denote the above gradient. Construct an alternative target $\tilde{\mathbf{x}}' = \tilde{\mathbf{x}} - \eta\tilde{\mathbf{u}}$ for some $\eta \in \mathbb{R}_+$. By standard gradient descent argument, there is some $\bar{\eta}$ such that any $\eta \in (0, \bar{\eta})$ yields an alternate target $\tilde{\mathbf{x}}'$ that improves over $\tilde{\mathbf{x}}$. ∎

We now prove that, controlling for scale, BMG can yield larger performance gains than MG. Recall that $\xi_G^\alpha(\mathbf{x}^{(K)}) = \mathbf{x}^{(K)} - \alpha G^T \nabla f \, \mathbf{x}^{(K)}$. Consider $\xi_G^r$, with $r := \|\nabla f(\mathbf{x}^{(K)})\|_2 / \|G^T \nabla f(\mathbf{x}^{(K)})\|_2$.

**Corollary 1.** *Let $\mu = \|\cdot\|_2^2$ and $\tilde{\mathbf{x}} = \xi_G^r(\mathbf{x}^{(K)})$. Let $\mathbf{w}'$ be given by Eq. 1 and $\tilde{\mathbf{w}}$ be given by Eq. 2. For $\beta$ sufficiently small, $f\big(\mathbf{x}^{(K)}(\tilde{\mathbf{w}})\big) \leq f\big(\mathbf{x}^{(K)}(\mathbf{w}')\big)$, with strict inequality if $GG^T \neq G^T$.*

*Proof.* Let $\mathbf{g} := \nabla_x f\big(\mathbf{x}^{(K)}\big)$. By Lemma 1, $f\big(\mathbf{x}^{(K)}(\mathbf{w}')\big) - f\big(\mathbf{x}^{(K)}(\mathbf{w})\big) = -\beta \langle G^T \mathbf{g}, \mathbf{g} \rangle + O(\beta^2)$. From Theorem 1, with $\mu = \|\cdot\|_2^2$, $f\big(\mathbf{x}^{(K)}(\tilde{\mathbf{w}})\big) - f\big(\mathbf{x}^{(K)}(\mathbf{w})\big) = -r \langle G^T \mathbf{g}, G^T \mathbf{g} \rangle + O(\beta(\alpha+\beta))$. For $\beta$ sufficiently small, the inner products dominate and we have

$$f\big(\mathbf{x}^{(K)}(\tilde{\mathbf{w}})\big) - f\big(\mathbf{x}^{(K)}(\mathbf{w}')\big) \approx -\beta \left( r \langle G^T \mathbf{g}, G^T \mathbf{g} \rangle - \langle G^T \mathbf{g}, \mathbf{g} \rangle \right).$$

To determine the sign of the expression in parenthesis, consider the problem

$$\max_{\mathbf{v} \in \mathbb{R}^{n_x}} \langle G^T \mathbf{g}, \mathbf{v} \rangle \qquad \text{s.t.} \quad \|\mathbf{v}\|_2 \leq 1.$$

Form the Lagrangian $\mathcal{L}(\mathbf{v}, \lambda) := \langle G^T \mathbf{g}, \mathbf{v} \rangle - \lambda(\|\mathbf{v}\|_2 - 1)$. Solve for first-order conditions:

$$G^T \mathbf{g} - \lambda \frac{\mathbf{v}^*}{\|\mathbf{v}^*\|_2} = 0 \implies \mathbf{v}^* = \frac{\|\mathbf{v}^*\|_2}{\lambda} G^T \mathbf{g}.$$

If $\lambda = 0$, then we must have $\|\mathbf{v}^*\|_2 0$, which clearly is not an optimal solution. Complementary slackness then implies $\|\mathbf{v}^*\|_2 = 1$, which gives $\lambda = \|\mathbf{v}^*\|_2 \|G^T \mathbf{g}\|_2$ and hence $\mathbf{v}^* = G^T \mathbf{g} / \|G^T \mathbf{g}\|_2$. By virtue of being the maximiser, $\mathbf{v}^*$ attains a higher function value than any other $\mathbf{v}$ with $\|\mathbf{v}\|_2 \leq 1$, in particular $\mathbf{v} = \mathbf{g} / \|\mathbf{g}\|_2$. Evaluating the objective at these two points gives

$$\frac{\langle G^T \mathbf{g}, G^T \mathbf{g} \rangle}{\|G^T \mathbf{g}\|_2} \geq \frac{\langle G^T \mathbf{g}, \mathbf{g} \rangle}{\|\mathbf{g}\|_2} \implies r \langle G^T \mathbf{g}, G^T \mathbf{g} \rangle \geq \langle G^T \mathbf{g}, \mathbf{g} \rangle,$$

where we use that $r = \|\mathbf{g}\|_2 / \|G^T \mathbf{g}\|_2$ by definition. Thus $f\big(\mathbf{x}^{(K)}(\tilde{\mathbf{w}})\big) \leq f\big(\mathbf{x}^{(K)}(\mathbf{w}')\big)$, with strict inequality if $GG^T \neq G^T$ and $G^T \mathbf{g} \neq \mathbf{0}$. ∎

## B  NON-STATIONARY NON-EPISODIC REINFORCEMENT LEARNING

### B.1  SETUP

This experiment is designed to provide a controlled setting to delineate the differences between standard meta-gradients and bootstrapped meta-gradients. The environment is a $5 \times 5$ grid world with two objects; a blue and a red square (Figure 7). Thus, we refer to this environment as the *two-colors* domain. At each step, the agent (green) can take an action to move either up, down, left, or right and observes the position of each square and itself. If the agent reaches a coloured square, it obtains a reward of either $+1$ or $-1$ while the colour is randomly moved to an unoccupied location. Every 100 000 steps, the reward for each object flips. For all other transitions, the agent obtains a reward of $-0.04$. Observations are constructed by concatenating one-hot encodings of the each $x$- and $y$-coordinate of the two colours and the agent's position, with a total dimension of $2 \times 3 \times 5 = 30$ (two coordinates for each of three objects, with each one-hot vector being 5-dimensional).

Figure 7: Two-colors Grid-world. The agent's goal is to collect either blue or red squared by navigating the green square.

The two-colors domain is designed such that the central component determining how well a memory-less agent adapts is its exploration. Our agents can only regulate exploration through policy entropy. Thus, to converge on optimal task behaviour, the agent must reduce policy entropy. Once the task switches, the agent encounters what is effectively a novel task (due to it being memory-less). To rapidly adapt the agent must first increase entropy in the policy to cover the state-space. Once the agent observe rewarding behaviour, it must then reduce entropy to converge on task-optimal behaviour.

All experiments run on the CPU of a single machine. The agent interacts with the environment and update its parameters synchronously in a single stream of experience. A *step* is thus comprised of the following operations, in order: (1) given observation, agent takes action, (2) if applicable, agent update its parameters, (3) environment transitions based on action and return new observation. The parameter update step is implemented differently depending on the agent, described below.

---

**Algorithm 1** $N$-step RL actor loop

---

**Require:** $N$                   ▷ Rollout length.
**Require:** $\mathbf{x} \in \mathbb{R}^{n_x}$                ▷ Policy parameters.
**Require:** $\mathbf{s}$                  ▷ Environment state.
 $\mathcal{B} \leftarrow (\mathbf{s})$                 ▷ Initialise rollout.
 **for** $t = 1, 2, \ldots, N$ **do**
  $\mathbf{a} \sim \pi_{\mathbf{x}}(\mathbf{s})$              ▷ Sample action.
  $\mathbf{s}, r \leftarrow \mathrm{env}(\mathbf{s}, \mathbf{a})$          ▷ Take a step in environment.
  $\mathcal{B} \leftarrow \mathcal{B} \cup (\mathbf{a}, r, \mathbf{s})$           ▷ Add to rollout.
 **end for**
 **return** $\mathbf{s}, \mathcal{B}$

---

**Algorithm 2** $K$-step online learning loop

---

**Require:** $N, K$          ▷ Rollout length, meta-update length.
**Require:** $\mathbf{x} \in \mathbb{R}^{n_x}, \mathbf{z} \in \mathbb{R}^{n_z}, \mathbf{w} \in \mathbb{R}^{n_w}$   ▷ Policy, value function, and meta parameters.
**Require:** $\mathbf{s}$                 ▷ Environment state.
 **for** $k = 1, 2, \ldots, K$ **do**
  $\mathbf{s}, \mathcal{B} \leftarrow \mathrm{ActorLoop}(\mathbf{x}, \mathbf{s}, N)$        ▷ Algorithm 1.
  $(\mathbf{x}, \mathbf{z}) \leftarrow \varphi((\mathbf{x}, \mathbf{z}), \mathcal{B}, \mathbf{w})$        ▷ Inner update step.
 **end for**
 **return** $\mathbf{s}, \mathbf{x}, \mathbf{z}, \mathcal{B}$

---

**Algorithm 3** Online RL with BMG

---

**Require:** $N, K, L$     ▷ Rollout length, meta-update length, bootstrap length.
**Require:** $\mathbf{x} \in \mathbb{R}^{n_x}, \mathbf{z} \in \mathbb{R}^{n_z}, \mathbf{w} \in \mathbb{R}^{n_w}$   ▷ Policy, value function, and meta parameters.
**Require:** $\mathbf{s}$                ▷ Environment state.
 $\mathbf{u} \leftarrow (\mathbf{x}, \mathbf{z})$
 **while** True **do**
  $\mathbf{s}, \mathbf{u}^{(K)}, \_ \leftarrow \mathrm{InnerLoop}(\mathbf{u}, \mathbf{w}, \mathbf{s}, N, K)$    ▷ $K$-step inner loop, Algorithm 2.
  $\mathbf{s}, \mathbf{u}^{(K+L-1)}, \mathcal{B} \leftarrow \mathrm{InnerLoop}(\mathbf{u}^{(K)}, \mathbf{w}, \mathbf{s}, N, L-1)$   ▷ $L-1$ bootstrap, Algorithm 2.
  $\tilde{\mathbf{u}} \leftarrow \mathbf{u}^{(K+L-1)} - \alpha \nabla_u \ell(\mathbf{u}^{(K+L-1)}, \mathcal{B})$    ▷ Gradient step on objective $\ell$.
  $\mathbf{w} \leftarrow \mathbf{w} - \beta \nabla_w \mu(\tilde{\mathbf{u}}, \mathbf{u}^{(K)}(\mathbf{w}))$      ▷ BMG outer step.
  $\mathbf{u} \leftarrow \mathbf{u}^{(K+L-1)}$        ▷ Continue from most resent parameters.
 **end while**

---

## B.2 Actor-Critic Experiments

**Agent** The first agent we evaluate is a simple actor-critic which implements a softmax policy ($\pi_{\mathbf{x}}$) and a critic ($v_{\mathbf{z}}$) using separate feed-forward MLPs. Agent parameter updates are done according to the actor-critic loss in Eq. 4 with the on-policy n-step return target. For a given parameterisation of the agent, we interact with the environment for $N = 16$ steps, collecting all observations, rewards, and actions into a rollout (Algorithm 1). When the rollout is full, the agent update its parameters under the actor-critic loss with SGD as the optimiser (Algorithm 2). To isolate the effect of meta-learning, all hyper-parameters except the entropy regularization weight ($\epsilon = \epsilon_{\mathrm{EN}}$) are fixed (Table 1); for each agent, we sweep for the learning rate that yields highest cumulative reward within a 10 million step budget. For the non-adaptive baseline, we additionally sweep for the best regularization weight.

**Meta-learning** To meta-learn the entropy regularization weight, we introduce a small MLP with meta-parameters $\mathbf{w}$ that ingests a statistic $\mathbf{t}$ of the learning process—the average reward over each of the 10 most recent rollouts—and predicts the entropy rate $\epsilon_{\mathbf{w}}(\mathbf{t}) \in \mathbb{R}_+$ to use in the agent's parameter update of $\mathbf{x}$. To compute meta-updates, for a given horizon $T = K$ or $T = K + (L-1)$, we fix $\mathbf{w}$ and make $T$ agent parameter updates to obtain a sequence $(\tau_1, \mathbf{x}^{(1)}, \mathbf{z}^{(1)}, \ldots, \tau_T, \mathbf{x}^{(T)}, \mathbf{z}^{(T)})$.

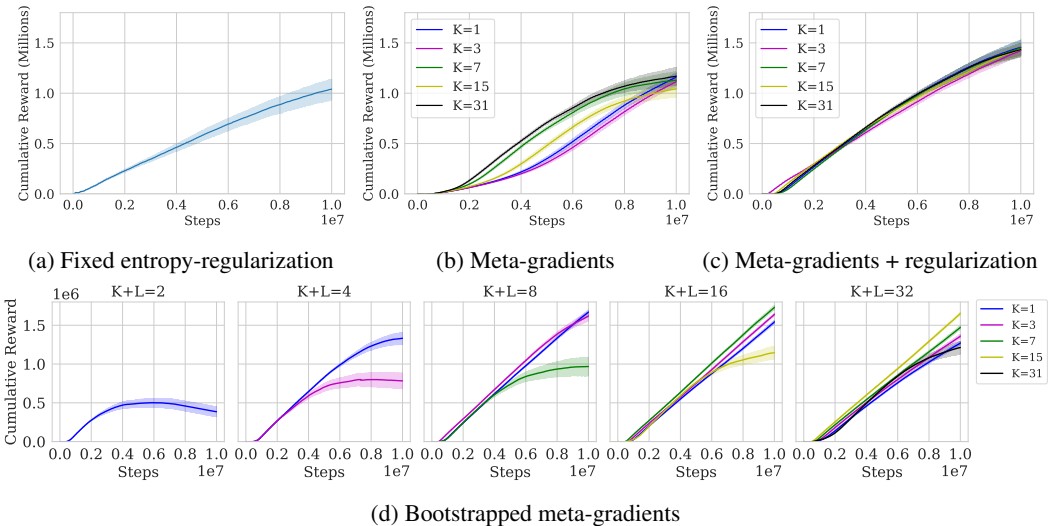

(a) Fixed entropy-regularization  (b) Meta-gradients  (c) Meta-gradients + regularization

(d) Bootstrapped meta-gradients

Figure 8: Total rewards on two-colors with actor-critics. Shading: standard deviation over 50 seeds.

**MG** is optimised by averaging each policy and entropy loss encountered in the sequence, i.e. the meta-objective is given by $\frac{1}{T} \sum_{t=1}^{T} \ell_{\text{PG}}^t(\mathbf{x}^{(t)}(\mathbf{w})) + \epsilon_{\text{meta}} \ell_{\text{EN}}^t(\mathbf{x}^{(t)}(\mathbf{w}))$, where $\epsilon_{\text{meta}} \in \{0, 0.1\}$ is a fixed hyper-parameter and $\ell^t$ implies that the objective is computed under $\tau_t$.

**BMG** is optimised by computing the matching loss $\mu_{\tau_T}(\tilde{\mathbf{x}}, \mathbf{x}^{(K)}(\mathbf{w}))$, where $\tilde{\mathbf{x}}$ is given by $\tilde{\mathbf{x}} = \mathbf{x}^{(T)} - \beta \nabla_x (\ell_{\text{PG}}^T(\mathbf{x}^{(T)}) + \epsilon_{\text{meta}} \ell_{\text{EN}}^T(\mathbf{x}^{(T)}))$. That is to say, the TB "unrolls" the meta-learner for $L-1$ steps, starting from $(\mathbf{x}^{(K)}, \mathbf{z}^{(K)})$, and takes a final policy-gradient step ($\epsilon_{\text{meta}} = 0$ unless otherwise noted). Thus, in this setting, our TB exploits that the first $(L-1)$ steps have already been taken by the agent during the course of learning (Algorithm 3). Moreover, the final $L$th step only differs in the entropy regularization weight, and can therefore be implemented without an extra gradient computation. As such, the meta-update under BMG exhibit no great computational overhead to the MG update. In practice, we observe no significant difference in wall-clock speed for a given $K$.

**Main experiment: detailed results** The purpose of our main experiment Section 5.1 is to (a) test whether larger meta-learning horizons—particularly by increasing $L$—can mitigate the short-horizon bias, and (b) test whether the agent can learn an exploration schedule without explicit domain knowledge in the meta-objective (in the form of entropy regularization). As reported in Section 5.1, we find the answer to be affirmative in both cases. To shed further light on these findings, Figure 8

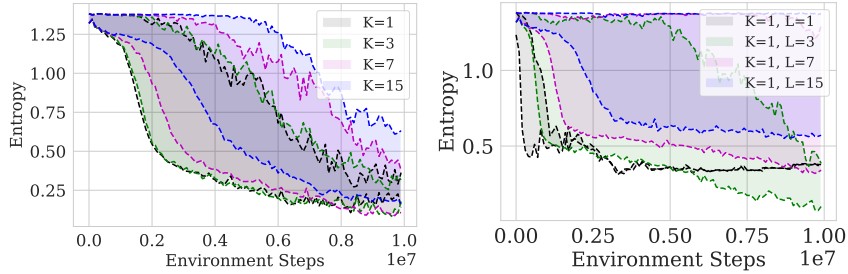

Figure 9: Range of the entropy of a softmax-policy over time (2-colors). Each shaded area shows the difference between the entropy 3333 steps after the agent observes a new entropy and the entropy after training on the reward-function for 100000 steps. Meta-gradients without explicit entropy-regularization (left) reduce entropy over time while Bootstrapped meta-gradients (right) maintain entropy with a large enough meta-learning horizon. Averaged across 50 seeds.

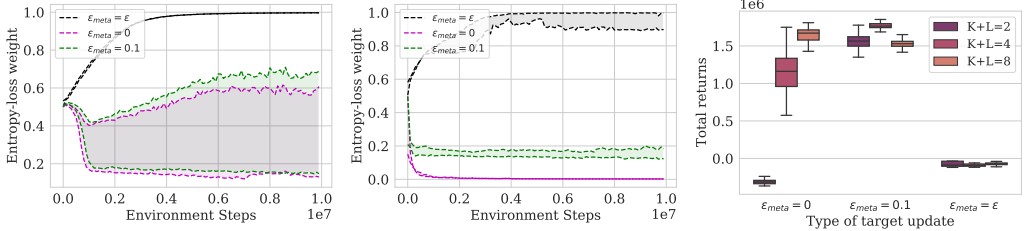

Figure 10: Ablations for actor-critic agent with BMG. Each shaded area shows the range of entropy regularization weights generated by the meta-learner. The range is computed as the difference between $\epsilon$ at the beginning and end of each reward-cycle. Left: entropy regularization weight range when $K = 1$ and $L = 7$. Center: entropy regularization weight range when $K = 1$ and $L = 1$. Right: For $K = 1$ effect of increasing $L$ with or without meta-entropy regularization. Result aggregated over 50 seeds.

reports cumulative reward curves for our main experiment in Section 5.1. We note that MG tends to collapse for any $K$ unless the meta-objective is explicitly regularized via $\epsilon_{\text{meta}}$. To characterise why MG fail for $\epsilon_{\text{meta}} = 0$, Figure 9 portrays the policy entropy range under either MG or BMG. MG is clearly overly myopic by continually shrinking the entropy range, ultimately resulting in a non-adaptive policy.

**Ablation: meta-regularization**  To fully control for the role of meta-regularization, we conduct further experiments by comparing BMG with and without entropy regularization (i.e. $\epsilon_{\text{meta}}$) in the $L$th target update step. Figure 10 demonstrates that BMG indeed suffers from myopia when $L = 1$, resulting in a collapse of the entropy regularization weight range. However, increasing the meta-learning horizon by setting $L = 7$ obtains a wide entropy regularization weight range. While adding meta-regularization does expand the range somewhat, the difference in total return is not statistically significant (right panel, Figure 10).

**Ablation: target bootstrap**  Our main TB takes $L - 1$ steps under the meta-learned update rule, i.e. the meta-learned entropy regularization weight schedule, and an $L$th policy-gradient step without entropy regularization. In this ablation, we very that taking a final step under a

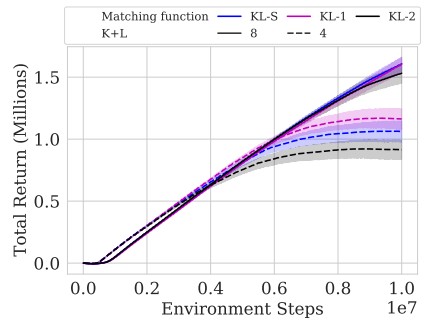

Figure 11: Total reward on two-colors with an actor-critic agent and different matching functions for BMG. Shading: standard deviation over 50 seeds.

*different* update rule is indeed critical. Figure 10 shows that, for $K = 1$ and $L \in \{1, 7\}$, using the meta-learned update rule for all target update steps leads to a positive feedback loop that results in maximal entropy regularization, leading to a catastrophic loss of performance (right panel, Figure 10).

**Ablation: matching function**  Finally, we control for different choices of matching function. Figure 11 contrasts the mode-covering version, KL-1, with the mode-seeking version, KL-2, as well as the symmetric KL. We observe that, in this experiment, this choice is not as significant as in other experiments. However, as in Atari, we find a the mode-covering version to perform slightly better.

### B.3  Q-LEARNING EXPERIMENTS

**Agent**  In this experiment, we test Peng's $Q(\lambda)$ (Peng & Williams, 1994) agent with $\varepsilon$-greedy exploration. The agent implements a feed-forward MLP to represent a Q-function $q_{\mathbf{x}}$ that is optimised online. Thus, agent parameter update steps do not use batching but is done online (i.e. on each step). To avoid instability, we use a momentum term that maintains an Exponentially Moving Average (EMA) over the agent parameter gradient. In this experiment we fix all hyper-parameters of the update rule (Table 1) and instead focuses on meta-learned $\varepsilon$-greedy exploration.

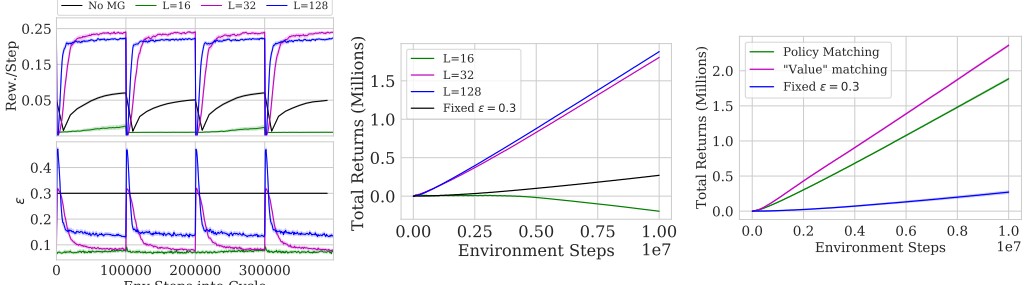

Figure 12: Results on two-colors under a $Q(\lambda)$ agent with meta-learned $\varepsilon$-greedy exploration under BMG. Averaged over 50 seeds.

**BMG**   We implement BMG in a similar fashion to the actor-critic case. The meta-learner is represented by a smaller MLP $\varepsilon_{\mathbf{w}}(\cdot)$ with meta-parameters $\mathbf{w}$ that ingests the last 50 rewards, denoted by $\mathbf{t}$, and outputs the $\varepsilon$ to use on the current time-step. That is to say, given meta-parameters $\mathbf{w}$, the agent's policy is defined by

$$\pi_{\mathbf{x}}(\mathbf{a} \mid \mathbf{s}_t, \mathbf{t}_t, \mathbf{w}) = \begin{cases} 1 - \varepsilon_{\mathbf{w}}(\mathbf{t}_t) + \frac{\varepsilon_{\mathbf{w}}(\mathbf{t}_t)}{|A|} & \text{if} \quad \mathbf{a} = \arg\max_{\mathbf{b}} q_{\mathbf{x}}(\mathbf{s}_t, \mathbf{b}) \\ \frac{\varepsilon_{\mathbf{w}}(\mathbf{t}_t)}{|A|} & \text{else.} \end{cases}$$

**Policy-matching**   This policy can be seen as a stochastic policy which takes the Q-maximizing action with probability $1 - \varepsilon$ and otherwise picks an action uniformly at random. The level of entropy in this policy is regulated by the meta-learner. We define a TB by defining a target policy under $q_{\tilde{\mathbf{x}}}$, where $\tilde{\mathbf{x}}$ is given by taking $L$ update steps. Since there are no meta-parameters in the update rule, all $L$ steps use the same update rule. However, we define the target policy as the greedy policy

$$\pi_{\tilde{\mathbf{x}}}(\mathbf{a} \mid \mathbf{s}_t) = \begin{cases} 1 & \text{if} \quad \mathbf{a} = \arg\max_{\mathbf{b}} q_{\tilde{\mathbf{x}}}(\mathbf{s}_t, \mathbf{b}) \\ 0 & \text{else.} \end{cases}$$

The resulting BMG update is simple: minimize the KL-divergence $\mu^{\pi}(\tilde{\mathbf{x}}, \mathbf{x}) \coloneqq \mathrm{KL}\left(\pi_{\tilde{\mathbf{x}}} \parallel \pi_{\mathbf{x}}\right)$ by adjusting the entropy in $\pi_{\mathbf{x}}$ through $\varepsilon_{\mathbf{w}}$. Thus, policy-matching under this target encourages the meta-learner to match a greedy policy-improvement operation on a target $q_{\tilde{\mathbf{x}}}$ that has been trained for a further $L$ steps. More specifically, if $\arg\max_{\mathbf{b}} q_{\tilde{\mathbf{x}}}(\mathbf{s}, \mathbf{b}) = \arg\max_{\mathbf{b}} q_{\mathbf{x}}(\mathbf{s}, \mathbf{b})$, so that the greedy policy improvement matches the target, then the matching loss is minimised by setting $\varepsilon = 0$. If greedy policy improvement does not correspond, so that acting greedily w.r.t. $q_{\mathbf{x}}$ does not match the target, then the matching loss is minimised by increasing entropy, i.e. increasing $\varepsilon$. The meta-objective is defined in terms of $\mathbf{x}$ as it does not require differentiation through the update-rule.

**'Value'-matching**   A disadvantage of policy matching is that it provides a sparse learning signal: $\varepsilon$ is increased when the target-policy differs from the current policy and decreased otherwise. The magnitude of the change depends solely on the current value of $\varepsilon$. It is therefore desirable to evaluate alternative matching functions that provide a richer signal. Inspired by value-matching for actor-critic agents, we construct a form of 'value' matching by taking the expectation over $q_{\mathbf{x}}$ under the induced stochastic policy, $u_{\mathbf{x}}(\mathbf{s}) \coloneqq \sum_{\mathbf{a} \in \mathcal{A}} \pi_{\mathbf{x}}(\mathbf{a} \mid \mathbf{s}) q_{\mathbf{x}}(\mathbf{s}, \mathbf{a})$. The resulting matching objective is given by

$$\mu^{u}(\tilde{\mathbf{x}}, \mathbf{x}) = \mathbb{E}\left[\left(u_{\tilde{\mathbf{x}}}(\mathbf{s}) - u_{\mathbf{x}}(\mathbf{s}; \mathbf{t}, \mathbf{w})\right)^2\right].$$

While the objective is structurally similar to value-matching, $u$ does not correspond to well-defined value-function since $q_{\mathbf{x}}$ is not an estimate of the action-value of $\pi_{\mathbf{x}}$.

**Detailed results**   Figure 12 shows the learned $\varepsilon$-schedules for different meta-learning horizons: if $L$ is large enough, the agent is able to increase exploration when the task switches and quickly recovers a near-optimal policy for the current cycle. Figure 12 further shows that a richer matching function, in this case in the form of 'value' matching, can yield improved performance.

Table 1: Two-colors hyper-parameters

| **Actor-critic** | |
| --- | --- |
| Inner Learner | |
| Optimiser | SGD |
| Learning rate | 0.1 |
| Batch size | 16 (losses are averaged) |
| $\gamma$ | 0.99 |
| $\mu$ | $\mathrm{KL}(\pi_{\tilde{x}}||\pi_{x'})$ |
| MLP hidden layers $(v, \pi)$ | 2 |
| MLP feature size $(v, \pi)$ | 256 |
| Activation Function | ReLU |
| | |
| Meta-learner | |
| Optimiser | Adam |
| $\epsilon$ (Adam) | $10^{-4}$ |
| $\beta_1, \beta_2$ | 0.9, 0.999 |
| Learning rate candidates | $\{3 \cdot 10^{-6}, 10^{-5}, 3 \cdot 10^{-5}, 10^{-4}, 3 \cdot 10^{-4}\}$ |
| MLP hidden layers $(\epsilon)$ | 1 |
| MLP feature size $(\epsilon)$ | 32 |
| Activation Function | ReLU |
| Output Activation | Sigmoid |

| $Q(\lambda)$ | |
| --- | --- |
| Inner Learner | |
| Optimiser | Adam |
| Learning Rate | $3 \cdot 10^{-5}$ |
| $\epsilon$ (Adam) | $10^{-4}$ |
| $\beta_1, \beta_2$ | 0.9, 0.999 |
| Gradient EMA | 0.9 |
| $\lambda$ | 0.7 |
| $\gamma$ | 0.99 |
| MLP hidden layers (Q) | 2 |
| MLP feature size (Q) | 256 |
| Activation Function | ReLU |
| | |
| Meta-learner | |
| Learning Rate | $10^{-4}$ |
| $\epsilon$ (Adam) | $10^{-4}$ |
| $\beta_1, \beta_2$ | 0.9, 0.999 |
| Gradient EMA | 0.9 |
| MLP hidden layers $(\epsilon)$ | 1 |
| MLP feature size $(\epsilon)$ | 32 |
| Activation Function | ReLU |
| Output Activation | Sigmoid |

## C  ATARI

**Setup**  Hyper-parameters are reported in Table 2. We follow the original IMPALA (Espeholt et al., 2018) setup, but do not down-sample or gray-scale frames from the environment. Following previous works (Xu et al., 2018; Zahavy et al., 2020), we treat each game level as a separate learning problem; the agent is randomly initialized at the start of each learning run and meta-learning is conducted online during learning on a single task, see Algorithm 6. We evaluate final performance between 190-200 million frames. All experiments are conducted with 3 independent runs under different seeds. Each of the 57 levels in the Atari suite is a unique environment with distinct visuals and game mechanics. Exploiting this independence, statistical tests of aggregate performance relies on a total sample size per agent of $3 \times 57 = 171$.

**Agent**  We use a standard feed-forward agent that received a stack of the 4 most recent frames (Mnih et al., 2013) and outputs a softmax action probability along with a value prediction. The agent is implemented as a deep neural network; we use the IMPALA network architecture without LSTMs, with larger convolution kernels to compensate for more a complex input space, and with a larger conv-to-linear projection. We add experience replay (as per (Schmitt et al., 2020)) to allow multiple steps on the target. All agents use the same number of online samples; unless otherwise stated, they also use the same number of replay samples. We ablate the role of replay data in Appendix C.2.

**STACX**  The IMPALA agent introduces specific form of importance sampling in the actor critic update and while STACX largely rely on the same importance sampling mechanism, it differs slightly to facilitate the meta-gradient flow. The actor-critic update in STACX is defined by Eq. 4 with the following definitions of $\rho$ and $G$. Let $\bar{\rho} \geq \bar{c} \in \mathbb{R}_+$ be given and let $\nu : \mathcal{S} \times \mathcal{A} \to [0, 1]$ represent the *behaviour policy* that generated the rollout. Given $\pi_{\mathbf{x}}$ and $v_{\bar{\mathbf{z}}}$, define the Leaky V-Trace target by

$$
\begin{aligned}
\eta_t &:= \pi_{\mathbf{x}}(\mathbf{a}_t \mid \mathbf{s}_t) / \nu(\mathbf{a}_t \mid \mathbf{s}_t) \\
\rho_t &:= \alpha_\rho \min\{\eta_t, \bar{\rho}\} + (1 - \alpha_\rho)\eta_t \\
c_i &:= \lambda \left( \alpha_c \min\{\eta_i, \bar{c}\} + (1 - \alpha_c)\eta_i \right) \\
\delta_t &:= \rho_t \left( \gamma v_{\bar{\mathbf{z}}}(\mathbf{s}_{t+1}) + r_{t+1} - v_{\bar{\mathbf{z}}}(\mathbf{s}_t) \right) \\
G_t^{(n)} &= v_{\bar{\mathbf{z}}}(\mathbf{s}_t) + \sum_{i=0}^{(n-1)} \gamma^i \left( \prod_{j=0}^{i-1} c_{t+j} \right) \delta_{t+i},
\end{aligned}
$$

with $\alpha_\rho \geq \alpha_c$. Note that—assuming $\bar{c} \geq 1$ and $\lambda = 1$—in the on-policy setting this reduces to the n-step return since $\eta_t = 1$, so $\rho_t = c_t = 1$. The original v-trace target sets $\alpha_\rho = \alpha_c = 1$.

STACX defines the main "task" as a tuple $(\pi^0, v^0, f(\cdot, \mathbf{w}_0))$, consisting of a policy, critic, and an actor-critic objective (Eq. 4) under Leaky V-trace correction with meta-parameters $\mathbf{w}_0$. Auxiliary tasks are analogously defined tuples $(\pi^i, v^i, f(\cdot, \mathbf{w}_i))$, $i \geq 1$. All policies and critics share the same feature extractor but differ in a separate MLP for each $\pi^i$ and $v^i$. The objectives differ in their hyper-parameters, with all hyper-parameters being meta-learned. Auxiliary policies are not used for acting; only the main policy $\pi^0$ interacts with the environment. The objective used to update the agent's parameters is the sum of all tasks (each task is weighted through $\epsilon_{\text{PG}}, \epsilon_{\text{EN}}, \epsilon_{\text{TD}}$). The objective used for the MG update is the original IMPALA objective under fixed hyper-parameters $\mathbf{p}$ (see Meta-Optimisation in Table 2). Updates to agent parameters and meta-parameters happen simultaneously on rollouts $\tau$. Concretely, let $\mathbf{m}$ denote parameters of the feature extractor, with $(\mathbf{x}_i, \mathbf{z}_i)$ denoting parameters of task $i$'s policy MLP and critic MLP. Let $\mathbf{u}_i := (\mathbf{m}, \mathbf{x}_i, \mathbf{z}_i)$ denote parameters of $(\pi^i, v^i)$, with $\mathbf{u} := (\mathbf{m}, \mathbf{x}_0, \mathbf{z}_0, \ldots \mathbf{x}_n, \mathbf{z}_n)$. Let $\mathbf{w} = (\mathbf{w}_0, \ldots, \mathbf{w}_n)$ and denote by $\mathbf{h}$ auxiliary vectors of the optimiser. Given (a batch of) rollout(s) $\tau$, the STACX update is given by

$$
\begin{aligned}
\left( \mathbf{u}^{(1)}, \mathbf{h}_u^{(1)} \right) &= \text{RMSProp}\left( \mathbf{u}, \mathbf{h}_u, \mathbf{g}_u \right) & \mathbf{g}_u &= \nabla_u \sum_{i=1}^{n} f_\tau\left( \mathbf{u}_i; \mathbf{w}_i \right) \\
\left( \mathbf{w}^{(1)}, \mathbf{h}_w^{(1)} \right) &= \text{Adam}\left( \mathbf{w}, \mathbf{h}_w, \mathbf{g}_w \right) & \mathbf{g}_w &= \nabla_w f_\tau\left( \mathbf{u}_0^{(1)}(\mathbf{w}); \mathbf{p} \right).
\end{aligned}
$$

**BMG**  We use the same setup, architecture, and hyper-parameters for BMG as for STACX unless otherwise noted; the central difference is the computation of $\mathbf{g}_w$. For $L = 1$, we compute the

bootstrapped meta-gradient under $\mu_\tau$ on data $\tau$ by

$$\mathbf{g}_w = \nabla_w \mu_\tau \left( \tilde{\mathbf{u}}_0, \mathbf{u}_0^{(1)}(\mathbf{w}) \right), \quad \text{where} \quad (\tilde{\mathbf{u}}_0, \_) = \text{RMSProp}\left( \mathbf{u}_0^{(1)}, \mathbf{h}_u^{(1)}, \nabla_u f_\tau\left( \mathbf{u}_0^{(1)}; \mathbf{p} \right) \right).$$

Note that the target uses the same gradient $\nabla_u f(\mathbf{u}_0^{(1)}; \mathbf{p})$ as the outer objective in STACX; hence, BMG does not use additional gradient information or additional data for $L = 1$. The *only* extra computation is the element-wise update required to compute $\tilde{\mathbf{u}}_0$ and the computation of the matching loss. We discuss computational considerations in Appendix C.4. For $L > 1$, we take $L - 1$ step under the meta-learned objective with different replay data in each update. To write this explicitly, let $\tau$ be the rollout data as above. Let $\tilde{\tau}^{(l)}$ denote a separate sample of only replay data used in the $l$th target update step. For $L > 1$, the TB is described by the process

$$\left( \tilde{\mathbf{u}}_0^{(1)}, \tilde{\mathbf{h}}_u^{(1)} \right) = \text{RMSProp}\left( \mathbf{u}_0^{(1)}, \mathbf{h}_u^{(1)}, \mathbf{g}_u^{(1)} \right), \qquad\qquad \mathbf{g}_u^{(1)} = \nabla_u \sum_{i=1}^n f_{\tilde{\tau}^{(1)}}\left( \mathbf{u}_i^{(1)}; \mathbf{w}_i \right)$$

$$\left( \tilde{\mathbf{u}}_0^{(2)}, \tilde{\mathbf{h}}_u^{(2)} \right) = \text{RMSProp}\left( \tilde{\mathbf{u}}_0^{(1)}, \tilde{\mathbf{h}}_u^{(1)}, \tilde{\mathbf{g}}_u^{(1)} \right), \qquad\qquad \tilde{\mathbf{g}}_u^{(1)} = \nabla_u \sum_{i=1}^n f_{\tilde{\tau}^{(2)}}\left( \tilde{\mathbf{u}}_i^{(1)}; \mathbf{w}_i \right)$$

$$\vdots$$

$$\left( \tilde{\mathbf{u}}_0, \_ \right) = \text{RMSProp}\left( \tilde{\mathbf{u}}_0^{(L-1)}, \tilde{\mathbf{h}}_u^{(L-1)}, \tilde{\mathbf{g}}_u^{(L-1)} \right), \qquad \tilde{\mathbf{g}}_u^{(L-1)} = \nabla_u f_\tau\left( \tilde{\mathbf{u}}_0^{(L-1)}, \mathbf{p} \right).$$

Targets and corresponding momentum vectors are discarded upon computing the meta-gradient. This TB corresponds to following the meta-learned update rule for $L - 1$ steps, with a final step under the IMPALA objective. We show in Appendix C.3 that this final step is crucial to stabilise meta-learning. For pseudo-code, see Algorithm 6.

Matching functions are defined in terms of the rollout $\tau$ and with targets defined in terms of the main task $\mathbf{u}_0$. Concretely, we define the following objectives:

$$\mu_\tau^\pi \left( \tilde{\mathbf{u}}_0, \mathbf{u}_0^{(1)}(\mathbf{w}) \right) = \text{KL}\left( \pi_{\tilde{\mathbf{u}}_0} \,\|\, \pi_{\mathbf{u}_0^{(1)}(\mathbf{w})} \right),$$

$$\mu_\tau^v \left( \tilde{\mathbf{u}}_0, \mathbf{u}_0^{(1)}(\mathbf{w}) \right) = \mathbb{E}\left[ \left( v_{\tilde{\mathbf{u}}_0} - v_{\mathbf{u}_0^{(1)}(\mathbf{w})} \right)^2 \right],$$

$$\mu_\tau^{\pi+v} \left( \tilde{\mathbf{u}}_0, \mathbf{u}_0^{(1)}(\mathbf{w}) \right) = \mu_\tau^\pi \left( \tilde{\mathbf{u}}_0, \mathbf{u}_0^{(1)}(\mathbf{w}) \right) + \lambda \mu_\tau^v \left( \tilde{\mathbf{u}}_0, \mathbf{u}_0^{(1)}(\mathbf{w}) \right), \qquad \lambda = 0.25,$$

$$\mu^{L2} \left( \tilde{\mathbf{u}}_0, \mathbf{u}_0^{(1)}(\mathbf{w}) \right) = \left\| \tilde{\mathbf{u}}_0 - \mathbf{u}_0^{(1)}(\mathbf{w}) \right\|_2.$$

---

**Algorithm 4** Distributed $N$-step RL actor loop

---

**Require:** $N$                                                      ▷ Rollout length.
**Require:** $\mathcal{R}$                               ▷ Centralised replay server.
**Require:** $d$                           ▷ Initial state method.
**Require:** $c$                     ▷ Parameter sync method.
  **while** True **do**
    **if** $|\mathcal{B}| = N$ **then**
      $\mathcal{R} \leftarrow \mathcal{R} \cup \mathcal{B}$      ▷ Send rollout to replay.
      $\mathbf{x} \leftarrow c()$      ▷ Sync parameters from learner.
      $\mathbf{s} \leftarrow d(\mathbf{s})$      ▷ Optional state reset.
      $\mathcal{B} \leftarrow (\mathbf{s})$      ▷ Initialise rollout.
    **end if**
    $\mathbf{a} \sim \pi_{\mathbf{x}}(\mathbf{s})$      ▷ Sample action.
    $\mathbf{s}, r \leftarrow \text{env}(\mathbf{s}, \mathbf{a})$      ▷ Take a step in environment.
    $\mathcal{B} \leftarrow \mathcal{B} \cup (\mathbf{a}, r, \mathbf{s})$      ▷ Add to rollout.
  **end while**

---

**Algorithm 5** $K$-step distributed learning loop

---

**Require:** $\mathcal{B}_1, \mathcal{B}_2, \ldots, \mathcal{B}_K$      ▷ $K$ $N$-step rollouts.
**Require:** $\mathbf{x} \in \mathbb{R}^{n_x}, \mathbf{z} \in \mathbb{R}^{n_z}, \mathbf{w} \in \mathbb{R}^{n_w}$      ▷ Policy, value function, and meta parameters.
  **for** $k = 1, 2, \ldots, K$ **do**
    $(\mathbf{x}, \mathbf{z}) \leftarrow \varphi((\mathbf{x}, \mathbf{z}), \mathcal{B}_k, \mathbf{w})$      ▷ Inner update step.
  **end for**
  **return** $\mathbf{x}, \mathbf{z}$

---

**Algorithm 6** Distributed RL with BMG

---

**Require:** $N, K, L, M$      ▷ Rollout length, meta-update length, bootstrap length, parallel actors.
**Require:** $\mathbf{x} \in \mathbb{R}^{n_x}, \mathbf{z} \in \mathbb{R}^{n_z}, \mathbf{w} \in \mathbb{R}^{n_w}$      ▷ Policy, value function, and meta parameters.
  $\mathbf{u} \leftarrow (\mathbf{x}, \mathbf{z})$
  Initialise $\mathcal{R}$ replay buffer      ▷ Collects $N$-step trajectories $\mathcal{B}$ from actors.
  Initialise $M$ asynchronous actors      ▷ Run concurrently, Algorithm 4.
  **while** True **do**
    $\{\mathcal{B}^{(k)}\}_{k=1}^{K+L} \sim \mathcal{R}$      ▷ Sample $K$ rollouts from replay.
    $\mathbf{u}^{(K)} \leftarrow \text{InnerLoop}(\mathbf{u}, \mathbf{w}, \{\mathcal{B}^{(k)}\}_{k=1}^{K})$      ▷ $K$-step inner loop, Algorithm 5.
    $\mathbf{u}^{(K+L-1)} \leftarrow \text{InnerLoop}(\mathbf{u}^{(K)}, \mathbf{w}, \{\mathcal{B}^{(l)}\}_{l=K}^{L-1})$      ▷ $L-1$-step bootstrap, Algorithm 5.
    $\tilde{\mathbf{u}} \leftarrow \mathbf{u}^{(K+L-1)} - \alpha \nabla_u \ell(\mathbf{u}^{(K+L-1)}, \mathcal{B}^{(K+L)})$      ▷ Gradient step on objective $\ell$.
    $\mathbf{w} \leftarrow \mathbf{w} - \beta \nabla_w \mu(\tilde{\mathbf{u}}, \mathbf{u}^{(K)}(\mathbf{w}))$      ▷ BMG outer step.
    $\mathbf{u} \leftarrow \mathbf{u}^{K}$      ▷ Optional: continue from $K + L - 1$ update.
    Send parameters $\mathbf{x}$ from learner to actors.
  **end while**

---

Table 2: Atari hyper-parameters

ALE (Bellemare et al., 2013)

| | |
|---|---|
| Frame dimensions (H, W, D) | 160, 210, 3 |
| Frame pooling | None |
| Frame grayscaling | None |
| Num. stacked frames | 4 |
| Num. action repeats | 4 |
| Sticky actions (Machado et al., 2018) | False |
| Reward clipping | $[-1, 1]$ |
| $\gamma = 0$ loss of life | True |
| Max episode length | 108 000 frames |
| Initial noop actions | 30 |

IMPALA Network (Espeholt et al., 2018)

| | |
|---|---|
| Convolutional layers | 4 |
| Channel depths | $64, 128, 128, 64$ |
| Kernel size | 3 |
| Kernel stride | 1 |
| Pool size | 3 |
| Pool stride | 2 |
| Padding | 'SAME' |
| Residual blocks per layer | 2 |
| Conv-to-linear feature size | 512 |

STACX (Zahavy et al., 2020)

| | |
|---|---|
| Auxiliary tasks | 2 |
| MLP hidden layers | 2 |
| MLP feature size | 256 |
| Max entropy loss value | 0.9 |

Optimisation

| | |
|---|---|
| Unroll length | 20 |
| Batch size | 18 |
| *of which from replay* | *12* |
| *of which is online data* | *6* |
| Replay buffer size | 10 000 |
| LASER (Schmitt et al., 2020) KL-threshold | 2 |
| Optimiser | RMSProp |
| Initial learning rate | $10^{-4}$ |
| Learning rate decay interval | 200 000 frames |
| Learning rate decay rate | Linear to 0 |
| Momentum decay | 0.99 |
| Epsilon | $10^{-4}$ |
| Gradient clipping, max norm | 0.3 |

Meta-Optimisation

| | |
|---|---|
| $\gamma, \lambda, \bar{\rho}, \bar{c}, \alpha$ | 0.995, 1, 1, 1, 1 |
| $\epsilon_{PG}, \epsilon_{EN}, \epsilon_{TD}$ | 1, 0.01, 0.25 |
| Optimiser | Adam |
| Learning rate | $10^{-3}$ |
| $\beta_1, \beta_2$ | 0.9, 0.999 |
| Epsilon | $10^{-4}$ |
| Gradient clipping, max norm | 0.3 |

## C.1 BMG Decomposition

In this section, we decompose the BMG agent to understand where observed gains come from. To do so, we begin by noting that—by virtue of Eq. 3—STACX is a special case of BMG under $\mu(\tilde{\mathbf{u}}, \mathbf{u}_0^{(1)}(\mathbf{w})) = \|\tilde{\mathbf{u}} - \mathbf{u}_0^{(1)}(\mathbf{w})\|_2^2$ with $\tilde{\mathbf{u}} = \mathbf{u}_0^{(1)} - \frac{1}{2}\nabla_u f_\tau(\mathbf{u}_0^{(1)}; \mathbf{p})$. That is to say, if the target is generated by a pure SGD step and the matching function is the squared L2 objective. We will refer to this configurations as SGD, L2. From this baseline—i.e. STACX—a minimal change is to retain the matching function but use RMSProp to generate the target. We refer t o this configuration as RMS, L2. From Corollary 1, we should suspect that correcting for curvature should improve performance. While RMSProp is not a representation of the metric $G$ in the analysis, it is nevertheless providing some form of curvature correction. The matching function can then be used for further corrections.

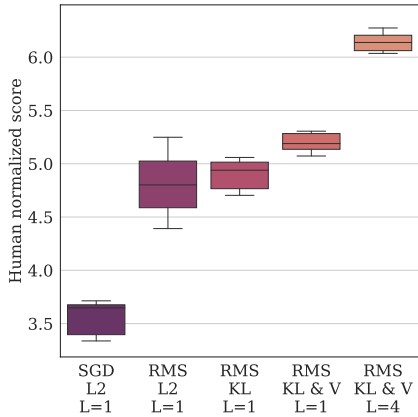

Figure 13: Atari BMG decomposition. We report human normalized score (median, quantiles, $\frac{1}{2}$IQR) between 190-200M frames over all 57 games, with 3 independent runs for each configuration.

Figure 13 shows that changing the target update rule from SGD to RMSProp, thereby correcting for curvature, yields a substantial gain. This supports our main claim that BMG can control for curvature and thereby facilitate meta-optimisation. Using the squared Euclidean distance in parameter space (akin to (Nichol et al., 2018; Flennerhag et al., 2019)) is surprisingly effective. However, it exhibits substantial volatility and is prone to crashing (c.f. Figure 15); changing the matching function to policy KL-divergence stabilizes meta-optimisation. Pure policy-matching leaves the role of the critic—i.e. policy evaluation—implicit. Having an accurate value function approximation is important to obtain high-quality policy gradients. It is therefore unsurprising that adding value matching provides a statistically significant improvement. Finally, we find that BMG can also mitigate myopia by extending the meta-learning horizon, in our TB by unrolling the meta-learned update rule for $L - 1$ steps. This is roughly as important as correcting for curvature, in terms of the relative performance gain.

To further support these findings, we estimate the effect BMG has on ill-conditioning and meta-gradient variance on three games where both STACX and BMG exhibit stable learning (to avoid confounding factors of non-stationary dynamics): Kangaroo, Star Gunner, and Ms Pacman. While

Table 3: Meta-gradient cosine similarity and variance per-game at 50-150M frames over 3 seeds.

|  | KL | KL & V | L2 | STACX |
|---|---|---|---|---|
| **Kangaroo** | | | | |
| Cosine similarity | 0.19 (0.02) | 0.11 (0.01) | 0.001 (1e-4) | 0.009 (0.01) |
| Meta-gradient variance | 0.05 (0.01) | 0.002 (1e-4) | 2.3e-9 (4e-9) | 6.4e-4 (7e-4) |
| Meta-gradient norm variance | 49 | 68 | 47 | 44 |
| **Ms Pacman** | | | | |
| Cosine similarity | 0.11 (0.006) | 0.03 (0.006) | 0.002 (4e-4) | -0.005 (0.01) |
| Meta-gradient variance | 90 (12) | 0.8 (0.2) | 9.6e-7 (2e-8) | 0.9 (0.2) |
| Meta-gradient norm variance | 2.1 | 7.9 | 4.2 | 2.1 |
| **Star Gunner** | | | | |
| Cosine similarity | 0.13 (0.008) | 0.07 (0.001) | 0.003 (5e-4) | 0.002 (0.02) |
| Meta-gradient variance | 4.2 (1.1) | 1.5 (2.3) | 1.9e-7 (3e-7) | 0.06 (0.03) |
| Meta-gradient norm variance | 6.1 | 6.6 | 11.7 | 6.5 |

the Hessian of the meta-gradient is intractable, an immediate effect of ill-conditioning is gradient interference, which we can estimate through cosine similarity between consecutive meta-gradients. We estimate meta-gradient variance on a per-batch basis. Table 3 presents mean statistics between 50M and 150M frames, with standard deviation over 3 seeds. BMG achieves a meta-gradient cosine similarity that is generally 2 orders of magnitude larger than that of STACX. It also explicitly demonstrates that using the KL divergence as matching function results in better curvature relative to using the L2 distance. The variance of the meta-gradient is larger for BMG than for STACX (under KL). This is due to intrinsically different gradient magnitudes. To make comparisons, we report the gradient norm to gradient variance ratio, which roughly indicates signal to noise. We note that in this metric, BMG tends to be on par with or lower than that of STACX.

## C.2   EFFECT OF REPLAY

We find that extending the meta-learning horizon by taking more steps on the target leads to large performance improvements. To obtain these improvements, we find that it is critical to re-sample replay data for each step, as opposed to re-using the same data for each rollout. Figure 14 demonstrates this for $L = 4$ on MsPacman. This can be explained by noting that reusing data allows the target to overfit to the current batch. By re-sampling replay data we obtain a more faithful simulation of what the meta-learned update rule would produce in $L - 1$ steps.

The amount of replay data is a confounding factor in the *meta-objective*. We stress that the agent parameter update is *always* the same in any experiment we run. That is to say, the additional use of replay data *only* affects the computation of the meta-objective. To control for this additional data in the meta-objective, we consider a subset of games where we see large improvements from $L > 1$. We run STACX and BMG with $L = 1$, but increase the amount of replay data used to compute the *meta-objective* to match the total amount of replay data used in the meta-objective when $L = 4$. This changes the online-to-replay ratio from $6 : 12$ to $6 : 48$ in the meta objective.

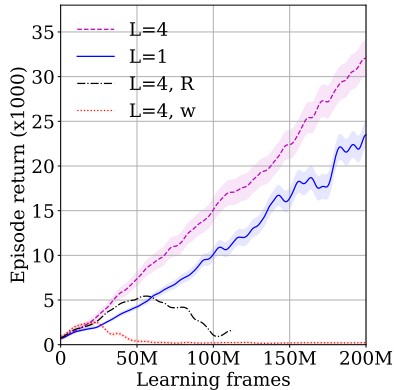

Figure 14: Atari, learning curves on MS Pacman for KL &V. $L = 4, R$ computes the $L$the step on only replay data. $L = 4, w$ uses the meta-learned objective for the $L$th step (with $L$th step computed on online and replay data, as per default). Shading depicts standard deviation across 3 seeds.

Figure 15 shows that the additional replay data is not responsible for the performance improvements we see for $L = 4$. In fact, we find that increasing the amount of replay data in the meta-objective exacerbates off-policy issues and leads to *reduced* performance. It is striking that BMG can make use of this extra off-policy data. Recall that we use *only* off-policy replay data to take the first $L - 1$ steps on the target, and use the original online-to-replay ratio ($6 : 12$) in the $L$th step. In Figure 14, we test the effect of using *only* replay for all $L$ steps and find that having online data in the $L$th update step is critical. These results indicate that BMG can make effective use of replay by simulating the effect of the meta-learned update rule on off-policy data and correct for potential bias using online data.

## C.3   L VS K

Given that increasing $L$ yields substantial gains in performance, it is interesting to compare against increasing $K$, the number of agent parameter updates to backpropagate through. For fair comparison, we use an identical setup as for $L > 1$, in the sense that we use new replay data for each of the initial $K - 1$ steps, while we use the default rollout $\tau$ for the $K$th step. Hence, the data characteristics for $K > 1$ are identical to those of $L > 1$.

However, an important difference arise because each update step takes $K$ steps on the agent's parameters. This means that—withing the 200 million frames budget, $K > 1$ has a computational advantage as it is able to do more updates to the agent's parameters. With that said, these additional $K - 1$ updates use replay data only.

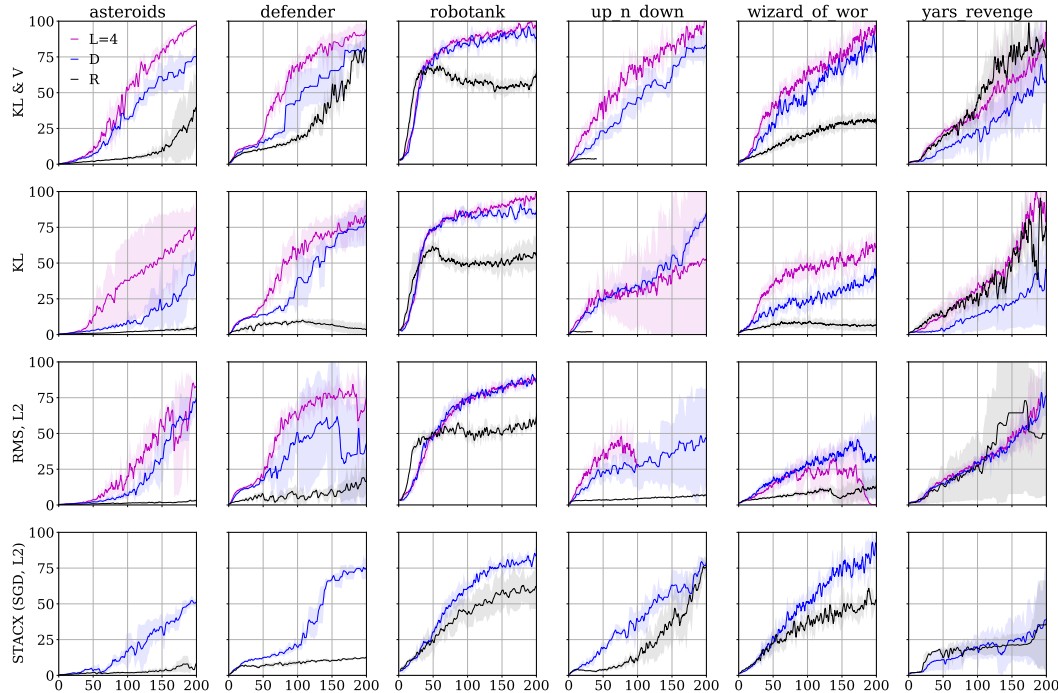

Figure 15: Atari experience replay ablation. We report episode returns, normalized to be in the range [0, max return] for each game for ease of comparison. Shading depicts standard deviation across 3 seeds. $D$ denotes default BMG configuration for $L = 1$, with $L = 4$ analgously defined. $R$ denotes $L = 1$, but with additional replay in the meta-objective to match the amount of replay used in $L = 4$.

Figure 16 demonstrates that increasing $K$ is fundamentally different from increasing $L$. We generally observe a loss of performance, again due to interference from replay. This suggests that target bootstrapping allows a fundamentally different way of extending the meta-learning horizon. In particular, these results suggests that meta-bootstrapping allows us to use relatively poor-quality (as evidence by $K > 1$) approximations to long-term consequences of the meta-learned update rule without impairing the agent's actual parameter update. Finally, there are substantial computational gains from increasing the meta-learning horizon via $L$ rather than $K$ (Figure 17).

## C.4 COMPUTATIONAL CHARACTERISTICS

IMPALA's distributed setup is implemented on a single machine with 56 CPU cores and 8 TPU (Jouppi et al., 2017) cores. 2 TPU cores are used to act in 48 environments asynchronously in parallel, sending rollouts to a replay buffer that a centralized learner use to update agent parameters and meta-parameters. Gradient computations are distributed along the batch dimension across the remaining 6 TPU cores. All Atari experiments use this setup; training for 200 millions frames takes 24 hours.

Figure 17 describes the computational properties of STACX and BMG as a function of the number of agent parameters and the meta-learning horizon, $H$. For STACX, the meta-learning horizon is defined by the number of update steps to backpropagate through, $K$. For BMG, we test one version which holds $L = 1$ fixed and varies $K$, as in for STACX, and one version which holds $K = 1$ ficed and varies $L$. To control for network size, we vary the number of channels in the convolutions of the network. We use a base of channels per layer, $x = (16, 32, 32, 16)$, that we multiply by a factor $1, 2, 4$. Thus we consider networks with kernel channels $1x = (16, 32, 32, 16)$, $2x = (32, 64, 64, 32)$, and $4x = (64, 128, 128, 64)$. Our main agent uses a network size (Table 2) equal to $4x$. We found that larger networks would not fit into memory when $K > 1$.

First, consider the effect of increasing $K$ (with $L = 1$ for BMG). For the small network ($1x$), BMG is roughly on par with STACX for all values of $K$ considered. However, BMG exhibits poorer scaling

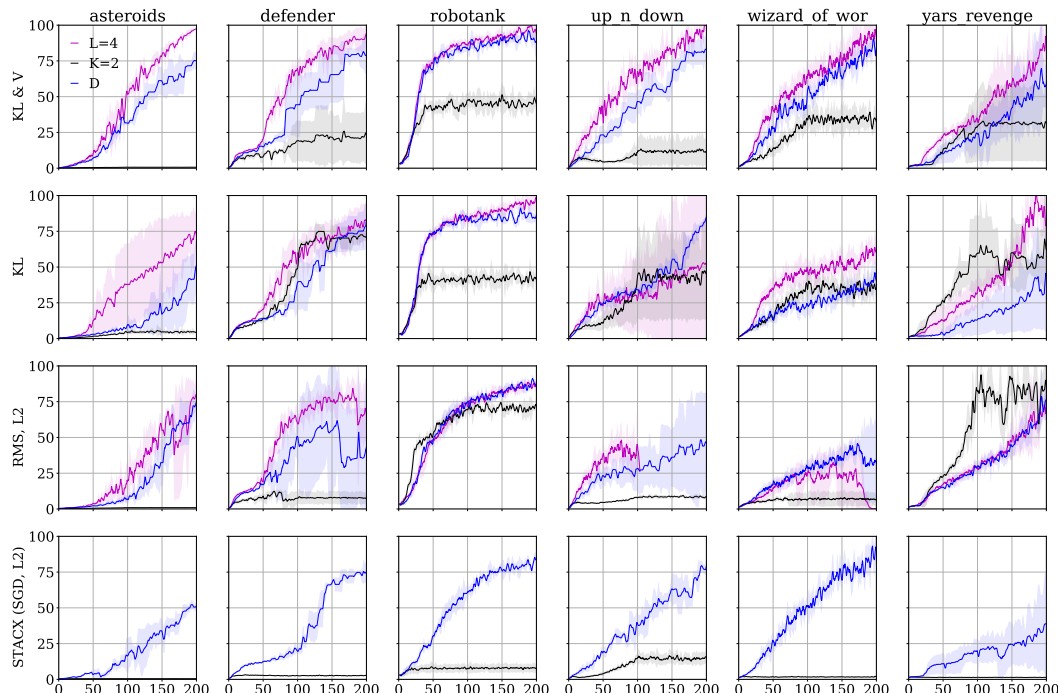

Figure 16: Atari $K$ vs $L$ ablation. We report episode returns, normalized to be in the range $[0, \text{max return}]$ for each game for ease of comparison. Shading depicts standard deviation across 3 seeds. $D$ denotes default BMG configuration for $L = 1$, with $L = 4$ analogously defined. $K = 2$ denotes $L = 1$, but $K = 2$ steps on agent parameters.

in network size, owing to the additional update step required to compute the target bootstrap. For $4x$, our main network configuration, we find that BMG is 20% slower in terms of wall-clock time. Further, we find that neither STACX nor BMG can fit the $4x$ network size in memory when $K = 8$.

Second, consider the effect of increasing $L$ with BMG (with $K = 1$). For $1x$, we observe no difference in speed for any $H$. However, increasing $L$ exhibits a dramatic improvement in scaling for $H > 2$—especially for larger networks. In fact, $L = 4$ exhibits a factor 2 speed-up compared to STACX for $H = 4, 4x$ and is two orders of magnitude faster for $H = 8, 2x$.

## C.5 ADDITIONAL RESULTS

Figure 19 presents per-game results learning curve for main configurations considered in this paper. Table 9 presents mean episode returns per game between 190-200 millions frames for all main

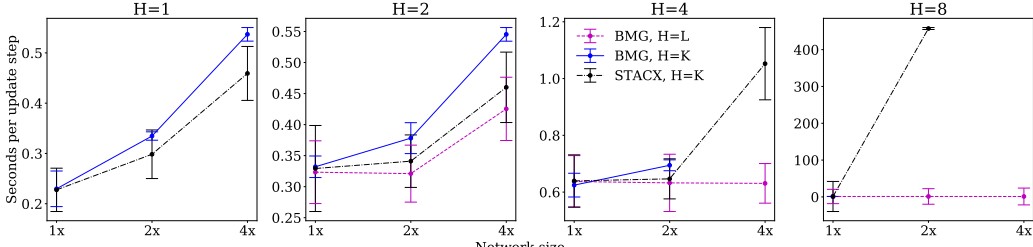

Figure 17: Atari: Computational characteristics as a function of network size (see Appendix C.4) and meta-learning horizon $H$. When $H = K$, we vary the number of update steps to backpropagate through (with $L = 1$ for BMG). When $H = L$, we vary the number of target update steps (with $K = 1$). Measurements are taken over the first 20 million learning frames on the game Pong.

configurations. Finally, we consider two variations of BMG in the $L = 1$ regime (Figure 18); one version (NS) re-computes the agent update after updating meta-parameters in a form of trust-region method. The other version (DB) exploits that the target has a taken a further update step and uses the target as new agent parameters. While NS is largely on par, interestingly, DB fails completely.

### C.6 DATA AND HYPER-PARAMETER SELECTION

We use the ALE Atari environment, publicly available at https://github.com/mgbellemare/Arcade-Learning-Environment, licensed under GNU GPL 2.0. Environment hyper-parameters were selected based on prior works (Mnih et al., 2013; Espeholt et al., 2018; Zahavy et al., 2020; Schmitt et al., 2020). Network, optimisation and meta-optimisation hyper-parameters are based on the original STACX implementation and tuned for optimal performance. Our median human normalized score matches published results. For BMG, we did not tune these hyper-parameters, except for $L > 1$. In this case, we observed that unique replay data in the initial $L - 1$ steps was necessary to yield any benefits. We observed a tendency to crash, and thus reduced the gradient clipping ratio from .3 to .2. For BMG configurations that use both policy and value matching, we tuned the weight on value matching by a grid search over $\{0.25, 0.5, 0.75\}$ on Ms Pacman, Zaxxon, Wizard of Wor, and Seaquest, with $0.25$ performing best.

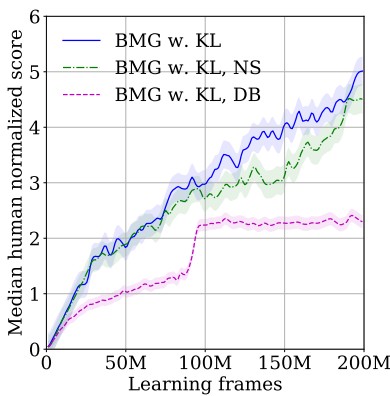

Figure 18: Atari BMG, alternative meta-update strategies. NS re-computes the agent-update the meta-update, akin to a trust-region method. DB uses the bootstrap target as the next agent parameters. Shading depicts standard deviation across 3 seeds.

## D MULTI-TASK META-LEARNING

### D.1 PROBLEM FORMULATION

Let $p(\tau)$ denote a given task distribution, where $\tau \in \mathbb{N}$ indexes a task $f^\tau$. Each task is also associated with distinct learner states $\mathbf{h}_\tau$ and task parameters $\mathbf{x}_\tau$, but all task learners use the same meta-learned update rule defined by meta-parameters $\mathbf{w}$. Hence, the meta-learner's problem is again to learn an update rule, but now in expectation over all learning problems. The MG update (Eq. 1) thus takes the form $\mathbf{w}' = \mathbf{w} - \beta \nabla_w \mathbb{E}_\tau [f^\tau(\mathbf{x}_\tau^{(K)}(\mathbf{w}))]$, where the expectation is with respect to $(f^\tau, \mathbf{h}_\tau, \mathbf{x}_\tau)$ and $\mathbf{x}_\tau^{(K)}(\mathbf{w})$ is the $K$-step update on task $\tau$ given $(f^\tau, \mathbf{h}_\tau, \mathbf{x}_\tau)$. Since $p(\tau)$ is independent of $\mathbf{w}$, this update becomes $\mathbf{w}' = \mathbf{w} - \beta \mathbb{E}_\tau [\nabla_w f^\tau(\mathbf{x}_\tau^{(K)}(\mathbf{w}))]$, i.e. the single-task meta-gradient in Section 3 in expectation over the task distribution.

With that said, the expectation involves integrating over $(\mathbf{h}_\tau, \mathbf{x}_\tau)$. This distribution is defined differently depending on the problem setup. In few-shot learning, $\mathbf{x}_\tau$ and $\mathbf{h}_\tau$ are typically a shared initialisations (Finn et al., 2017; Nichol et al., 2018; Flennerhag et al., 2019) and $f^\tau$ differ in terms of the data (Vinyals et al., 2016). However, it is possible to view the expectation as a prior distribution over task parameters (Grant et al., 2018; Flennerhag et al., 2020). In online multi-task learning, this expectation often reduces to an expectation over current task-learning states (Rusu et al., 2015; Denevi et al., 2019).

The BMG update is analogously defined. Given a TB $\xi$, define the task-specific target $\tilde{\mathbf{x}}_\tau$ given $\mathbf{x}_\tau^{(K)}$ by $\xi(\mathbf{x}_\tau^{(K)})$. The BMG meta-loss takes the form $\mathbf{w}' = \mathbf{w} - \beta \nabla_w \mathbb{E}_\tau [\mu_\tau(\tilde{\mathbf{x}}_\tau, \mathbf{x}_\tau^{(K)}(\mathbf{w}))]$, where $\mu_\tau$ is defined on data from task $\tau$. As with the MG update, as the task distribution is independent of $\mathbf{w}$, this simplifies to $\mathbf{w}' = \mathbf{w} - \beta \mathbb{E}_\tau [\nabla_w \mu_\tau(\tilde{\mathbf{x}}_\tau, \mathbf{x}_\tau^{(K)}(\mathbf{w}))]$, where $\mu_\tau$ is the matching loss defined on task data from $\tau$. Hence, as with MG, the multi-task BMG update is an expectation over the single-task BMG update in Section 3. See Algorithm 7 for a detailed description.

## D.2 FEW-SHOT MINIIMAGENET

**Setup** MiniImagenet (Vinyals et al., 2016; Ravi & Larochelle, 2017) is a sub-sample of the Imagenet dataset (Deng et al., 2009). Specifically, it is a subset of 100 classes sampled randomly from the 1000 classes in the ILSVRC-12 training set, with 600 images for each class. We follow the standard protocol (Ravi & Larochelle, 2017) and split classes into a non-overlapping meta-training, meta-validation, and meta-tests sets with 64, 16, and 20 classes in each, respectively. The datasset is licenced under the MIT licence and the ILSVRC licence. The dataset can be obtained from `https://paperswithcode.com/dataset/miniimagenet-1`. $M$-shot-$N$-way classification tasks are sampled following standard protocol (Vinyals et al., 2016). For each task, $M = 5$ classes are randomly sampled from the train, validation, or test set, respectively. For each class, $K$ observations are randomly sampled without replacement. The task validation set is constructed similarly from a disjoint set of $L = 5$ images per class. We follow the original MAML protocol for meta-training (Finn et al., 2017), taking $K$ task adaptation steps during meta-training and 10 adaptation steps during meta testing.

We study how the data-efficiency and computational efficiency of the BMG meta-objective compares against that of the MG meta-objective. To this end, for data efficiency, we report the meta-test set performance as we vary the number of meta-batches each algorithm is allow for meta-training. As more meta-batches mean more meta-tasks, this metric captures how well they leverage additional data. For computational efficiency, we instead report meta-test set performace as a function of total meta-training time. This metric captures computational trade-offs that arise in either method.

For any computational budget in either regime (i.e. $N$ meta-batches or $T$ hours of training), we report meta-test set performance across 3 seeds for the hyper-configuration with best validation performance (Table 4). This reflects the typical protocol for selecting hyper-parameters, and what each method would attain under a given budget. For both methods, we sweep over the meta-learning rate $\beta$; for shorter training runs, a higher meta-learning is critical to quickly converge. This however lead to sub-optimal performance for larger meta-training budgets, where a smaller meta-learning rate can produce better results. The main determinant for computational cost is the number of steps to backpropagate through, $K$. For BMG, we sweep over $K \in \{1, 5\}$. For MG, we sweep over $K \in \{1, 5, 10\}$. We allow $K = 10$ for MAML to ensure fair comparison, as BMG can extend its effective meta-learning horizon through the target bootstrap; we sweep over $L \in \{1, 5, 10\}$. Note that the combination of $K$ and $L$ effectively lets BMG interpolate between different computational trade-offs. Standard MG does not have this property, but several first-order approximations have been proposed: we allow the MG approach to switch from a full meta-gradient to either the FOMAML approximation (Finn et al., 2017) or the ANIL approximation (Raghu et al., 2020).

**Model, compute, and shared hyper-parameters** We use the standard convolutional model (Vinyals et al., 2016), which is a 4-layer convolutional model followed by a final linear layer. Each convolutional layer is defined by a $3 \times 3$ kernel with 32 channels, strides of 1, with batch normalisation, a ReLU activation and $2 \times 2$ max-pooling. We use the same hyper-parameters of optimisation and meta-optimisation as in the original MAML implementation except as specified in Table 4. Each model is trained on a single machine and runs on a V100 NVIDIA GPU.

Table 4: Hyper-parameter sweep per computational budget.

|  | MAML | BMG |
|---|---|---|
| $\beta$ | $\{0.0001, 0.001\}$ | $\{0.0001, 0.001\}$ |
| $K$ | $\{1, 5, 10\}$ | $\{1, 5\}$ |
| $L$ | — | $\{1, 5, 10\}$ |
| $\mu$ | — | $\{\text{KL}\left(\tilde{\mathbf{x}} \,\|\, \cdot\right), \text{KL}\left(\cdot \,\|\, \tilde{\mathbf{x}}\right)\}$ |
| FOMAML | { True, False } | — |
| ANIL | { True, False } | — |
| Total | 24 | 24 |

Table 6: Effect of BMG on ill-conditioning and meta-gradient variance on 5-way-5-shot MiniImagenet. Estimated meta-gradient cosine similarity ($\theta$) between consecutive gradients, meta-gradient variance ($\mathbb{V}$), and meta-gradient norm to variance ratio ($\rho$). Standard deviation across 5 independent seeds.

| $K$ | $L$ | MAML $\theta$ | $\mathbb{V}$ | $\rho$ | BMG $\theta$ | $\mathbb{V}$ | $\rho$ |
|---|---|---|---|---|---|---|---|
| 1 | 1 | 0.17 (0.01) | 0.21 (0.01) | 0.02 (0.02) | 0.17 (0.01) | 0.0002 (5e-6) | 0.59 (0.03) |
|   | 5 | | | | 0.18 (0.01) | 0.001 (1e-5) | 0.23 (0.01) |
|   | 10 | | | | 0.19 (0.01) | 0.0003 (2e-5) | 0.36 (0.01) |
| 5 | 1 | 0.03 (0.01) | 0.07 (0.009) | 0.08 (0.03) | 0.03 (0.005) | 0.01 (9e-5) | 0.84 (0.03 |
|   | 5 | | | | 0.04 (0.005) | 0.001 (5e-5) | 0.46 (0.02) |
|   | 10 | | | | 0.05 (0.004) | 0.003 (3e-5) | 0.18 (0.02) |

## D.3 ANALYSIS

In terms of data-efficiency, Table 7 reports best hyper-parameters for each data budget. For both BMG and MG, we note that small budgets rely on fewer steps to backpropagate through and a higher learning rate. BMG tends to prefer a higher target bootstrap in this regime. MG switches to backpropagation through $K > 1$ sooner than BMG, roughly around 70 000 meta-updates, while BMG switches around 120 000 meta-updates. This accounts for why BMG can achieve higher performance faster, as it can achieve similar performance without backpropagating through more than one update. It is worth noting that as BMG is given larger training budgets, to prevent meta-overfitting, shorter target bootstraps generalize better. We find that other hyper-parameters are not important for overall performance.

Table 5: Meta-training steps per second for MAML and BMG on 5-way-5-shot MiniImagenet. Standard deviation across 5 seeds in parenthesis.

| $K$ | $L$ | $H = K + L$ | MAML | BMG |
|---|---|---|---|---|
| 1 | 1 | 2 | 14.3 (0.4) | 12.4 (0.5) |
|   | 5 | 6 | - | 6.9 (0.3) |
|   | 10 | 11 | - | 4.4 (0.1) |
| 5 | 1 | 6 | 4.4 (0.06) | 4.2 (0.04) |
|   | 5 | 10 | - | 3.2 (0.03) |
|   | 10 | 15 | - | 2.5 (0.01) |
| 10 | 1 | 11 | 2.3 (0.01) | 2.2 (0.01) |
|   | 5 | 15 | - | 1.9 (0.01) |
|   | 10 | 20 | - | 1.7 (0.01) |
| 15 | - | 15 | 1.4 (0.01) | - |
| 20 | - | 20 | 1.1 (0.01) | - |

In terms of computational efficiency, Table 7 reports best hyper-parameters for each time budget. The pattern here follows a similar trend. MG does btter under a lower learning rate already after 4 hours, whereas BMG switches after about 8 hours. This data highlights the dominant role $K$ plays in determining training time.

We compare wall-clock time per meta-training step for various values of $K$ and $L$ Table 5. In our main configuration, i.e. $K = 5, L = 10$, BMG achieves a throughput of 2.5 meta-training steps per second, compared to 4.4 for MAML, making BMG 50% slower. In this setting, BMG has an effective meta-learning horizon of 15, whereas MAML has a horizon of 5. For MAML to achieve an effective horizon of 15, it's throughput would be reduced to 1.4, instead making MAML 56% slower than BMG.

Finally, we conduct a similar analysis as on Atari (Appendix C.1) to study the effect BMG has on ill-conditioning and meta-gradient variance. We estimate ill-conditioning through cosine similarity between consecutive meta-gradients, and meta-gradient variance on a per meta-batch basis. We report mean statistics for the 5-way-5-shot setup between 100 000 and 150 000 meta-gradient steps, with standard deviation over 5 independent seeds, in Table 6.

Unsurprisingly, MAML and BMG are similar in terms of curvature, as both can have a KL-divergence type of meta-objective. BMG obtains greater cosine similarity as $L$ increases, suggesting that BMG can transfer more information by having a higher temperature in its target. However, BMG exhibits substantially lower meta-gradient variance, and the ratio of meta-gradient norm to variance is an order of magnitude larger.

---

**Algorithm 7** Supervised multi-task meta-learning with BMG

---

**Require:** $K, L$        ▷ meta-update length, bootstrap length
**Require:** $M, N, T$        ▷ meta-batch size, inner batch size, meta-training steps.
**Require:** $\mathbf{x} \in \mathbb{R}^{n_x}, \mathbf{w} \in \mathbb{R}^{n_w}$        ▷ model and meta parameters.
  **for** $t = 1, 2, \ldots, T$ **do**
      $\mathbf{g} \leftarrow 0$        ▷ Initialise meta-gradient.
      **for** $i = 1, 2, \ldots, M$ **do**
         $\tau \sim p(\tau)$        ▷ Sample task.
         $\mathbf{x}_\tau \leftarrow \mathbf{x}$        ▷ For MAML, set $\mathbf{x} = \mathbf{w}$.
         **for** $k = 1, 2, \ldots, K$ **do**
            $\boldsymbol{\zeta}_\tau \sim p_{\text{train}}(\boldsymbol{\zeta} \mid \tau)$        ▷ Sample batch of task training data.
            $\mathbf{x}_\tau = \mathbf{x}_\tau + \varphi(\mathbf{x}_\tau, \boldsymbol{\zeta}_\tau, \mathbf{w})$        ▷ Task adaptation.
         **end for**
         $\mathbf{x}^{(K)} \leftarrow \mathbf{x}$        ▷ $K$-step adaptation.
         **for** $l = 1, 2, \ldots, L - 1$ **do**
            $\boldsymbol{\zeta}_\tau \sim p_{\text{test}}(\boldsymbol{\zeta} \mid \tau)$        ▷ Sample batch of task test data.
            $\mathbf{x}_\tau = \mathbf{x}_\tau + \varphi(\mathbf{x}_\tau, \boldsymbol{\zeta}_\tau, \mathbf{w})$        ▷ $L - 1$ step bootstrap.
         **end for**
         $\boldsymbol{\zeta}_\tau \sim p_{\text{test}}(\boldsymbol{\zeta} \mid \tau)$
         **if** final gradient step **then**        ▷ Assign target.
            $\tilde{\mathbf{x}}_\tau = \mathbf{x}_\tau - \alpha \nabla_x \ell(\mathbf{x}_\tau, \boldsymbol{\zeta}_\tau)$
         **else**
            $\tilde{\mathbf{x}}_\tau \leftarrow \mathbf{x}_\tau + \varphi(\mathbf{x}_\tau, \boldsymbol{\zeta}_\tau, \mathbf{w})$
         **end if**
         $\mathbf{g} \leftarrow \mathbf{g} + \nabla_w \mu(\tilde{\mathbf{x}}_\tau, \mathbf{x}^{(K)}(\mathbf{w}))$
      **end for**
      $\mathbf{w} \leftarrow \mathbf{w} - \frac{\beta}{M} \mathbf{g}$        ▷ BMG outer step.
  **end for**

---

Table 7: Data-efficiency: mean meta-test accuracy over 3 seeds for best hyper-parameters per data budget. $\mu = 1$ corresponds to KL $(\tilde{\mathbf{x}} \parallel \cdot)$ and $\mu = 2$ to KL $(\cdot \parallel \tilde{\mathbf{x}})$.

| Step (K) | $\beta$ | $K$ | $L$ | $\mu$ | Acc. (%) | $\beta$ | $K$ | FOMAML | ANIL | Acc. (%) |
|---|---|---|---|---|---|---|---|---|---|---|
| 10 | $10^{-3}$ | 1 | 10 | 1 | 61.4 | $10^{-3}$ | 1 | False | True | 61.7 |
| 20 | $10^{-3}$ | 1 | 10 | 1 | 61.8 | $10^{-3}$ | 1 | False | False | 61.9 |
| 30 | $10^{-3}$ | 1 | 10 | 1 | 62.5 | $10^{-3}$ | 10 | False | True | 62.3 |
| 40 | $10^{-3}$ | 5 | 1 | 1 | 63.1 | $10^{-3}$ | 5 | False | False | 62.7 |
| 50 | $10^{-3}$ | 5 | 1 | 1 | 63.5 | $10^{-3}$ | 10 | False | True | 62.9 |
| 60 | $10^{-3}$ | 5 | 1 | 1 | 63.7 | $10^{-3}$ | 1 | False | False | 63.0 |
| 70 | $10^{-3}$ | 1 | 1 | 2 | 63.7 | $10^{-3}$ | 5 | False | False | 63.0 |
| 80 | $10^{-3}$ | 5 | 1 | 1 | 63.7 | $10^{-4}$ | 5 | False | False | 63.1 |
| 90 | $10^{-3}$ | 5 | 1 | 1 | 63.8 | $10^{-3}$ | 5 | False | False | 63.3 |
| 100 | $10^{-3}$ | 1 | 1 | 2 | 63.8 | $10^{-4}$ | 5 | False | False | 63.4 |
| 110 | $10^{-3}$ | 1 | 1 | 2 | 63.9 | $10^{-4}$ | 5 | False | False | 63.6 |
| 120 | $10^{-4}$ | 5 | 5 | 1 | 63.9 | $10^{-4}$ | 5 | False | False | 63.6 |
| 130 | $10^{-4}$ | 5 | 10 | 1 | 64.0 | $10^{-4}$ | 5 | False | False | 63.6 |
| 140 | $10^{-4}$ | 5 | 5 | 1 | 64.1 | $10^{-4}$ | 5 | False | False | 63.6 |
| 150 | $10^{-4}$ | 5 | 5 | 1 | 64.2 | $10^{-4}$ | 10 | False | True | 63.6 |
| 160 | $10^{-4}$ | 5 | 5 | 1 | 64.3 | $10^{-4}$ | 5 | False | False | 63.6 |
| 170 | $10^{-4}$ | 5 | 5 | 1 | 64.4 | $10^{-4}$ | 5 | False | False | 63.7 |
| 180 | $10^{-4}$ | 5 | 5 | 1 | 64.5 | $10^{-4}$ | 10 | False | False | 63.8 |
| 190 | $10^{-4}$ | 5 | 10 | 1 | 64.6 | $10^{-4}$ | 5 | False | False | 63.9 |
| 200 | $10^{-3}$ | 5 | 10 | 2 | 64.7 | $10^{-4}$ | 10 | False | False | 64.0 |
| 210 | $10^{-4}$ | 5 | 1 | 1 | 64.7 | $10^{-4}$ | 5 | False | False | 64.1 |
| 220 | $10^{-4}$ | 5 | 5 | 1 | 64.7 | $10^{-4}$ | 10 | False | False | 64.2 |
| 230 | $10^{-4}$ | 5 | 5 | 1 | 64.8 | $10^{-4}$ | 5 | False | False | 64.2 |
| 240 | $10^{-4}$ | 5 | 1 | 2 | 64.8 | $10^{-4}$ | 5 | False | False | 64.1 |
| 250 | $10^{-4}$ | 5 | 5 | 1 | 64.9 | $10^{-4}$ | 5 | False | False | 64.1 |
| 260 | $10^{-4}$ | 5 | 1 | 1 | 64.9 | $10^{-4}$ | 5 | False | False | 64.0 |
| 270 | $10^{-4}$ | 5 | 1 | 1 | 64.8 | $10^{-4}$ | 5 | False | False | 63.9 |
| 280 | $10^{-4}$ | 5 | 1 | 1 | 64.8 | $10^{-4}$ | 5 | False | False | 63.8 |
| 290 | $10^{-4}$ | 5 | 1 | 1 | 64.7 | $10^{-4}$ | 5 | False | False | 63.8 |
| 300 | $10^{-4}$ | 5 | 5 | 1 | 64.7 | $10^{-4}$ | 5 | False | False | 63.8 |

Table 8: Computational-efficiency: mean meta-test accuracy over 3 seeds for best hyper-parameters per time budget. $\mu = 1$ corresponds to KL $(\tilde{\mathbf{x}} \parallel \cdot)$ and $\mu = 2$ to KL $(\cdot \parallel \tilde{\mathbf{x}})$.

| Time (h) | $\beta$ | $K$ | $L$ | $\mu$ | Acc. (%) | $\beta$ | $K$ | FOMAML | ANIL | Acc. (%) |
|---|---|---|---|---|---|---|---|---|---|---|
| 1 | $10^{-3}$ | 1 | 1 | 2 | 63.5 | $10^{-3}$ | 1 | False | False | 63.0 |
| 2 | $10^{-3}$ | 1 | 1 | 2 | 63.6 | $10^{-3}$ | 10 | False | True | 63.0 |
| 3 | $10^{-3}$ | 5 | 1 | 1 | 63.7 | $10^{-3}$ | 5 | False | False | 63.0 |
| 4 | $10^{-3}$ | 5 | 1 | 1 | 63.8 | $10^{-4}$ | 5 | False | True | 63.1 |
| 4 | $10^{-3}$ | 5 | 1 | 1 | 63.8 | $10^{-4}$ | 1 | False | True | 63.4 |
| 5 | $10^{-3}$ | 5 | 1 | 1 | 63.8 | $10^{-4}$ | 5 | False | False | 63.5 |
| 6 | $10^{-3}$ | 5 | 10 | 1 | 63.8 | $10^{-4}$ | 5 | False | False | 63.6 |
| 7 | $10^{-4}$ | 5 | 1 | 1 | 63.8 | $10^{-4}$ | 5 | False | False | 63.6 |
| 8 | $10^{-3}$ | 5 | 1 | 1 | 63.8 | $10^{-4}$ | 5 | False | False | 63.6 |
| 9 | $10^{-4}$ | 5 | 1 | 1 | 63.9 | $10^{-4}$ | 5 | False | False | 63.6 |
| 10 | $10^{-4}$ | 5 | 1 | 1 | 64.2 | $10^{-4}$ | 5 | False | False | 63.7 |
| 11 | $10^{-4}$ | 5 | 5 | 1 | 64.3 | $10^{-4}$ | 5 | False | False | 63.8 |
| 12 | $10^{-4}$ | 5 | 5 | 1 | 64.5 | $10^{-4}$ | 5 | False | False | 63.9 |
| 13 | $10^{-4}$ | 5 | 5 | 1 | 64.6 | $10^{-4}$ | 5 | False | False | 63.9 |
| 14 | $10^{-4}$ | 5 | 1 | 2 | 64.7 | $10^{-4}$ | 5 | False | False | 63.8 |
| 15 | $10^{-4}$ | 5 | 1 | 1 | 64.8 | $10^{-4}$ | 5 | False | False | 63.4 |
| 16 | $10^{-4}$ | 5 | 1 | 1 | 64.8 | $10^{-3}$ | 10 | False | False | 63.2 |
| 17 | $10^{-4}$ | 5 | 1 | 1 | 64.8 | $10^{-4}$ | 10 | False | False | 63.3 |
| 18 | $10^{-4}$ | 5 | 10 | 1 | 64.8 | $10^{-4}$ | 10 | False | False | 63.5 |
| 19 | $10^{-4}$ | 5 | 5 | 1 | 64.8 | $10^{-4}$ | 10 | False | False | 63.6 |
| 20 | $10^{-4}$ | 5 | 5 | 1 | 64.7 | $10^{-4}$ | 10 | False | False | 63.8 |
| 21 | $10^{-4}$ | 5 | 10 | 1 | 64.7 | $10^{-4}$ | 10 | False | False | 63.9 |
| 21 | $10^{-4}$ | 5 | 10 | 1 | 64.7 | $10^{-4}$ | 10 | False | False | 63.8 |
| 22 | $10^{-4}$ | 5 | 5 | 1 | 64.7 | $10^{-4}$ | 10 | False | False | 63.9 |
| 23 | $10^{-4}$ | 5 | 10 | 1 | 64.7 | $10^{-4}$ | 10 | False | False | 63.8 |
| 24 | $10^{-4}$ | 5 | 10 | 1 | 64.7 | — | — | — | — | — |

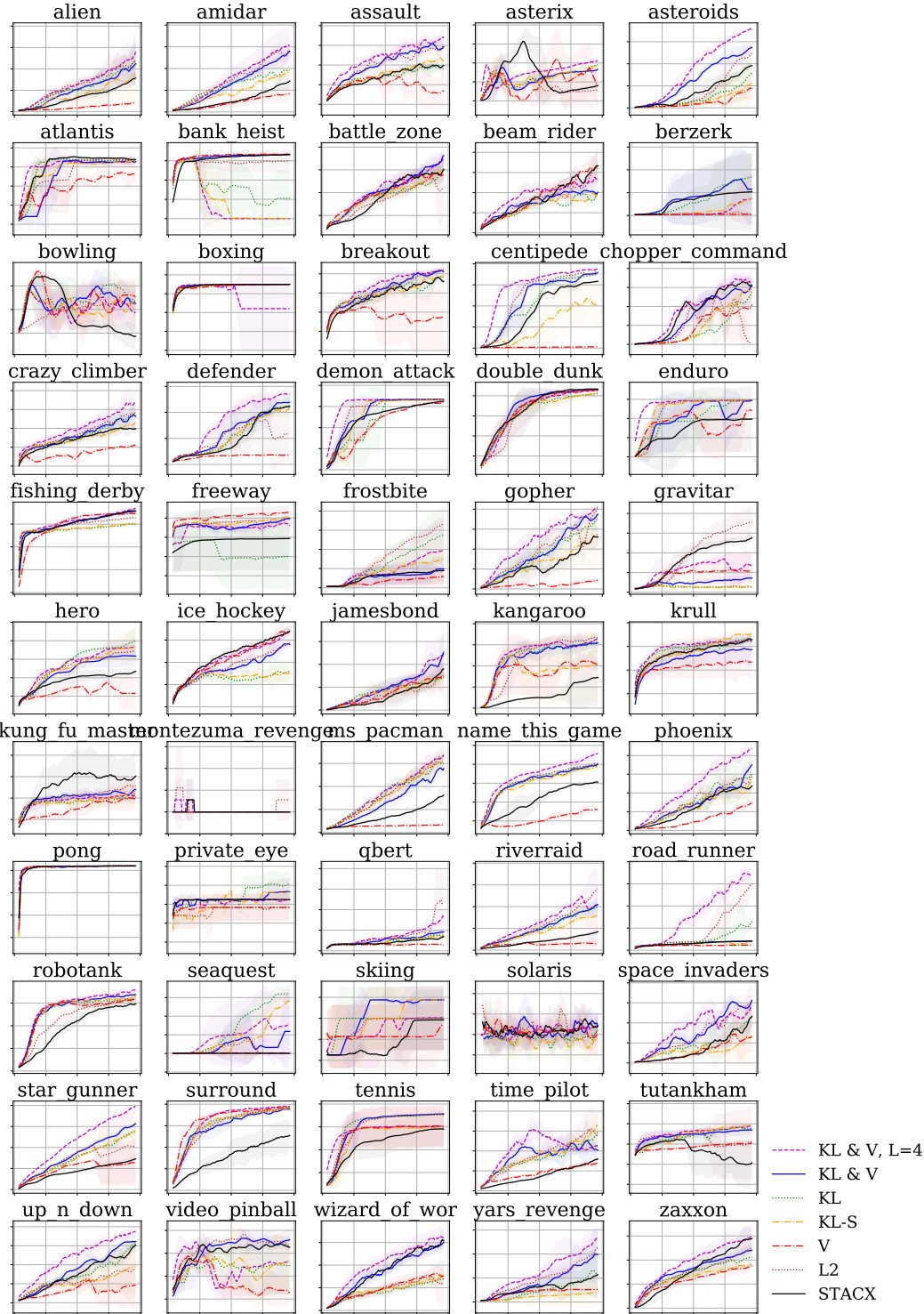

Figure 19: Atari, per-game performance across 3 seeds. Shading depicts standard deviation.

Table 9: Mean per-game performance between 190-200M frames.

| | KL | KL & V | KL & V, L=4 | KL-S | L2 | STACX | V |
|---|---|---|---|---|---|---|---|
| Alien | 45677 | 44880 | 58067 | 35750 | 50692 | 31809 | 7964 |
| Amidar | 4800 | 7099 | 7528 | 4974 | 7691 | 3719 | 1896 |
| Assault | 20334 | 29473 | 33019 | 21747 | 28301 | 19648 | 4101 |
| Asterix | 511550 | 439475 | 533385 | 487367 | 6798 | 245617 | 86053 |
| Asteroids | 145337 | 238320 | 289689 | 8585 | 220366 | 156096 | 56577 |
| Atlantis | 831920 | 813772 | 814780 | 806698 | 854441 | 848007 | 648988 |
| Bank Heist | 571 | 1325 | 0 | 13 | 1165 | 1329 | 1339 |
| Battle Zone | 73323 | 88407 | 88350 | 78941 | 50453 | 78359 | 72787 |
| Beam Rider | 37170 | 51649 | 57409 | 41454 | 67726 | 62892 | 74397 |
| Berzerk | 21146 | 2946 | 1588 | 2183 | 240 | 1523 | 1069 |
| Bowling | 46 | 50 | 42 | 46 | 50 | 28 | 52 |
| Boxing | 100 | 100 | 86 | 100 | 100 | 100 | 100 |
| Breakout | 742 | 832 | 847 | 774 | 827 | 717 | 16 |
| Centipede | 537032 | 542730 | 558849 | 291569 | 550394 | 478347 | 8895 |
| Chopper Command | 830772 | 934863 | 838090 | 736012 | 11274 | 846788 | 341350 |
| Crazy Climber | 233445 | 212229 | 265729 | 199150 | 229496 | 182617 | 126353 |
| Defender | 393457 | 374012 | 421894 | 364053 | 69193 | 344453 | 55152 |
| Demon Attack | 132508 | 133109 | 133571 | 132529 | 133469 | 130741 | 129863 |
| Double Dunk | 22 | 23 | 23 | 21 | 23 | 24 | 23 |
| Enduro | 2349 | 2349 | 2350 | 2360 | 2365 | 259 | 2187 |
| Fishing Derby | 41 | 63 | 68 | 41 | 52 | 62 | 59 |
| Freeway | 10 | 30 | 25 | 31 | 30 | 18 | 33 |
| Frostbite | 8820 | 3895 | 3995 | 5547 | 13477 | 2522 | 1669 |
| Gopher | 116010 | 116037 | 122459 | 92185 | 122790 | 87094 | 11920 |
| Gravitar | 271 | 709 | 748 | 259 | 3594 | 2746 | 944 |
| Hero | 60896 | 48551 | 52432 | 56044 | 51631 | 35559 | 20235 |
| Ice Hockey | 5 | 15 | 20 | 4 | 15 | 19 | 20 |
| Jamesbond | 22129 | 25951 | 30157 | 25766 | 18200 | 26123 | 23263 |
| Kangaroo | 12200 | 12557 | 13174 | 1940 | 13235 | 3182 | 8722 |
| Krull | 10750 | 9768 | 10510 | 11156 | 10502 | 10480 | 8899 |
| Kung Fu Master | 51038 | 58732 | 54354 | 54559 | 63632 | 67823 | 54584 |
| Montezuma Revenge | 0 | 0 | 0 | 0 | 0 | 0 | 0 |
| Ms Pacman | 25926 | 22876 | 28279 | 26267 | 27564 | 12647 | 2759 |
| Name This Game | 31203 | 31863 | 36838 | 30912 | 32344 | 24616 | 12583 |
| Phoenix | 529404 | 542998 | 658082 | 407520 | 440821 | 370270 | 247854 |
| Pitfall | 0 | -1 | 0 | 0 | 0 | 0 | 0 |
| Pong | 21 | 21 | 21 | 21 | 21 | 21 | 21 |
| Private Eye | 165 | 144 | 98 | 130 | 67 | 100 | 68 |
| Qbert | 87214 | 37135 | 72320 | 30047 | 75197 | 27264 | 3901 |
| Riverraid | 129515 | 132751 | 32300 | 91267 | 177127 | 47671 | 26418 |
| Road Runner | 240377 | 61710 | 521596 | 17002 | 424588 | 62191 | 34773 |
| Robotank | 64 | 66 | 71 | 65 | 64 | 61 | 65 |
| Seaquest | 684870 | 2189 | 82925 | 616738 | 1477 | 1744 | 3653 |
| Skiing | -10023 | -8988 | -9797 | -8988 | -9893 | -10504 | -13312 |
| Solaris | 2120 | 2182 | 2188 | 1858 | 2194 | 2326 | 2202 |
| Space Invaders | 35762 | 54046 | 40790 | 11314 | 49333 | 34875 | 15424 |
| Star Gunner | 588377 | 663477 | 790833 | 587411 | 39510 | 298448 | 43561 |
| Surround | 9 | 9 | 10 | 9 | 9 | 3 | 9 |
| Tennis | 23 | 24 | 23 | 21 | 24 | 19 | 24 |
| Time Pilot | 94746 | 60918 | 68626 | 95854 | 93466 | 49932 | 40127 |
| Tutankham | 282 | 268 | 291 | 280 | 288 | 101 | 205 |
| Up N Down | 342121 | 303741 | 381780 | 109392 | 202715 | 315588 | 17252 |
| Venture | 0 | 0 | 0 | 0 | 0 | 0 | 0 |
| Video Pinball | 230252 | 479861 | 399094 | 505212 | 485852 | 441220 | 77100 |
| Wizard Of Wor | 21597 | 45731 | 49806 | 22936 | 10817 | 47854 | 24250 |
| Yars Revenge | 77001 | 286734 | 408061 | 32031 | 398656 | 113651 | 77169 |
| Zaxxon | 44280 | 49448 | 59011 | 36261 | 49734 | 56952 | 35494 |

