# OpenReview forum: "Bootstrapped Meta-Learning"
_ICLR.cc/2022/Conference — ICLR 2022 Oral_

### Official Review · Reviewer_SXZx · 2021-11-02

**Correctness:** 4
**Technical Novelty And Significance:** 4
**Empirical Novelty And Significance:** 4
**Recommendation:** 8
**Confidence:** 3

**Main Review:**

**Strengths**

-  The paper is generally clear and well written.  It is content dense in and certain areas would benefit from more detailed discussions.  However, the appendices are very detailed and provide answers to almost all of the questions that arise during first reading.  In my suggestions below I’ve highlighted a few areas I feel this content would be especially beneficial (and one pare where the main text could be stripped-back as there is always a trade-off with fixed page limits).

- The key ideas of BMG, bootstrapping a target to combat myopic MG updates and using matching functions to improve the meta-learning dynamics, are novel and well motivated.  I strongly believe that these ideas will be of interest to the broader meta-learning community and will likely lead to multiple future research directions.

- The empirical results are strong and thorough.  The toy-model grid world in Sec 5.1 effectively highlights key properties of the proposed algorithm — such as the ability of BMG to exploit longer meta-learning horizons — and extensive testing on Atari environments (Sec 5.2) shows a significant improvement of STACX and leaves little doubt of the efficacy of BMG for self-tuning algorithms.   Moreover, the demonstrated performance improvement of BMG over MAML (sec 6) suggests broad possible applications and further endorses the proposed algorithm.

- BMG also has a couple of nice properties besides raw performance: (i) achieving less myopic updates without having to backpropagate through many update steps of the inner-loop parameters and (ii) allowing for the meta-learning of “behavioural parameters” that are not used in the learning rule (exploration epsilon is used as the example).  Whilst (i) is clearly desirable in the pursuit of computational efficiency, (ii) is very intriguing and opens the door to new applications of meta-learning (indeed, whilst it is mentioned in the abstract and summaries, I believe this point is slightly undersold in the main text and that this experiment could be presented in more detail than the single paragraph at the end of Sec 5.1 it is given — however, given the page limit I can understand the authors predicament).

**Weakenesses**

- My primary concern is not with the content of the paper, but that the it can be quite difficult to intuitively link the numerous experiments back to intuition or interpretation of the results.  I believe this was largely because the exact methodologies are difficult to follow and, indeed, I found it was essential to refer to the appendices repeatedly to fully unpack the experiments and appreciate the results.  Whilst I am sympathetic to space constraints, I believe the authors would be well served to provide more detailed descriptions in the main text or, ideally, an algorithm box.

- I find the implementation and implications of the experiment on multi-task few-shot learning (Sec 6) unclear in the following regards:
	- I do not understand the intuition of why a “hot” expert transforms more information than a “cold” expert, and, moreover, why BMG is able to use this to improve performance.  Could the authors clarify these points.
	- I feel this section in particular suffers from a lack of formal introduction of the task and methodology.  For example, the adaptation seems to be defined as a single step, whereas the appendix (D.2) notes that 10 are used during meta-testing.  Concretely, I think a formal description of the training procedure for BMG would be appropriate in the main text.

- I do not find the analysis presented in Sec 4 (“Performance Guarantees”) especially insightful.  My reading is that, whilst the MG update presented in Lemma 1 can guarantee local improvement, practical implementations of BMG do not.  Empirically, however, grounding the bootstrapped target with a single final step on the meta-objective is sufficient for good performance.  This conclusion is evident from the experiments themselves, and so I would have no issue if Sec 4 was in the appendix.

**Errata**

- Sec 3, paragraph starting “To tackle myopia…”: the inline equation for $\tilde{x}$ is missing brackets around the step counters for $x^{(K+L-1)}$.
- Sec 5.2, second last paragraph: typo - “K is more sensitive curvature and the quality of data”.
- Sec 6, sub-section BMG: typo - “raising the temperature in the expect allows”.
- Sec 6, sub-section Setup: typo - “For 50 meta-updates and beyond..”: this should be 50k meta-updates I believe.

**Summary Of The Paper:**

The paper presents Bootstrapped meta-gradients (BMG), an extension of typical meta-gradients (MG) for the task tuning meta-parameters that control the learning process (i.e. update step) of a learner.  In general terms, MG applies a (meta-)parameterised update rule to a learner for K steps, and then backpropagates through these updates to update the meta-parameters in the direction that improves the performance of the adapted learner.  The authors identify two limitations to this approach, and propose BMG to address them. (i) MG is myopic, in the sense that it does not account for future learning dynamics beyond these K steps, therefore BMG proposes to bootstrap a target from the K-step parameters (in practice by continuing to optimise w.r.t. parameterised update rule for L-1 steps, and then taking a final step w.r.t. a fixed objective to ground the signal). (ii) MG updates are necessarily restricted to be within the geometry of the parameterised learning process.  In contrast, BMG introduces a matching function to measure the distance between the learners K-step parameters and bootstrapped target in an arbitrary (and hopefully more-suitable space).

After framing the problem and BMG, the authors provide a discussion of the necessary conditions for BMG to guarantee performance improvements, though ultimately the presented algorithm is justified empirically using experiments on (i) a toy RL problem, (ii) the Atari RL test suite, (iii) multi-task few-shot adaptation on an image recognition task.  In all settings, BMG provides significant improvement over meta-learning baselines — most impressively achieving a new SOTA on Atari.  Moreover, key features of BMG are highlighted, including the ability to extend the meta-learning horizon without increasing the number of updates steps through which we must backpropagate, and that behavioural parameters outside of the update rule (specifically, epsilon in epsilon-greedy exploration) can also be meta-learned.

**Summary Of The Review:**

Overall I believe this paper proposed an insightful and novel approach to addressing the stated limitations of typical MG approaches.  The authors do a good job of motivating the key innovations proposed and, given the significant research interest in MG’s in the past few years, it is reasonable to assume that the methodologies presented will be of broad interest.  Moreover, the experimental results are impressive and clearly demonstrate the improved performance of BMG and analyse where this comes from.  I do believe that the experimental results would be better served with more detailed descriptions of the problem settings and methodologies, however given the overall level of detail presented in the appendices I do not doubt there validity or significance.  Even so, for the stated reasons of novelty, interest and potential impact, I believe the paper is suitable for acceptance in the current form.

---

> ### Author Response · Authors · 2021-11-23
> **Thank you for your review**
>
>
> Dear reviewer,
>
> Thank you for a thorough and in-depth review that will help us strengthen the manuscript. We will take your suggestions into account as we prepare the final version of the manuscript; please see below for answers to your questions.
>
> - *Main text vs appendix*. Thank you for bringing this to our attention; we will do a careful review of the paper as we prepare the final version and see if we can add more experimental details in the main paper, as well as adding algorithm boxes for each of the experimental setup.
>
> - *Few-shot experiment*. “Hot expert” vs “Cold expert”. The intuition follows from the distillation argument [1]: that is, by raising the temperature of the target distribution, the resulting “[...] soft targets have high entropy, they provide much more information per training case than hard targets [from a cold expert] and much less variance in the gradient between training cases.”
>
> - *Formal definition of training procedure*. Thank you for your suggestion, we will provide a more detailed description of the training procedure in the final version of the paper.
>
> - *Theoretical results*. Thank you for sharing your thoughts. We have strengthened Thm 1 to apply to a wider class of target bootstraps. To say something more precise than what we have in the paper, further assumptions on the objective, update, and matching function are needed; we hope that the analysis we presented can serve as inspiration for further theoretical research into more specific problem settings.
>
>  [1] Hinton, Vinyals & Dean: Distilling the Knowledge in a Neural Network. 2015.

---

### Official Review · Reviewer_N6Le · 2021-11-02

**Correctness:** 4
**Technical Novelty And Significance:** 3
**Empirical Novelty And Significance:** 3
**Recommendation:** 10
**Confidence:** 5

**Main Review:**

Strengths
- The proposed idea is simple.
- The writing is clear.
- To my knowledge, the proposed idea is novel yet concretely linked to many past works by virtue of generalizing them. The authors give the precise form to recover MG from BMG.
- The proposed idea results in strong improvement of the STACX agent in Atari-57, resulting in a state-of-the-art result (caveat: for model-free agents; the gap to model-based agents remains large).
- The authors demonstrate that the proposed idea is suitable in few-shot image classification, a popular application of meta-learning for few-shot learning.
- The authors run informative ablation studies that support the intuition behind the benefit of BMG: resolving curvature and mitigating myopia.

Weaknesses
- Given that you say BMG is compatible with any update function (so long as it is differentiable in the meta-parameters), it would be nice to have some experiments on learned sequence model update rules (e.g. RNN). All current experiments use update rules with a fixed functional form.
- I am not putting much weight on section 4 ("Performance Guarantees") given the gap between its assumptions and results vs. what is actually implemented, and the restriction to local optimization.

Minor comments
- p. 3, target bootstrap paragraph, 3rd line: are we missing a learning rate for the expression of the target?
- p. 5, the actor-critic RL objective and Eq. 4: as-is, we are always minimizing policy entropy; is there a sign error?

**Summary Of The Paper:**

This paper broadly considers meta-learning, a.k.a. bilevel optimization, across single-task, multi-task, supervised learning, and reinforcement learning settings. The authors aim to resolve two issues with the standard outer-loop gradient-based optimization of the meta-parameters (assuming a differentiable inner-loop): first, since the meta-learning objective is typically computed from learner parameters after applying up to $K$ inner-loop updates, the meta-optimization is myopic in that it does not optimize for further inner-loop improvement after $K$ steps; second, since the functional form of the learner's objective $f$ is used to drive the outer-loop updates, the meta-learning objective inherits the curvature of $f$. The main algorithmic contribution consists of a family of meta-learning objectives called bootstrapped meta-learning, in which meta-parameters are optimized to bring post-inner-loop learner parameters $x^{(K)}$ closer to a bootstrap target (which are also learner parameters) computed from $x^{(K)}$. The authors show that bootstrapped meta-learning generalizes the "direct" (my terminology) gradient-based meta-parameter optimization used in many previous works, recovering it when using specific choices for the bootstrap computation function and learner parameter matching function. With certain strong assumptions, the authors theoretically motivate the use of gradient-based bootstrap target functions for bootstrapped meta-learning in terms of optimization progress. The authors make several experimental contributions: they use bootstrapped meta-learning to achieve state-of-the-art model-free performance on Atari-57, demonstrate the viability of bootstrapped meta-learning in few-shot image classification on miniImageNet, and show that the more flexible, general form of bootstrapped meta-learning can enable meta-learning parameters that do not appear in the computation graph for the task objective, e.g. meta-learning the exploration rate of the behavior policy in $\epsilon$-greedy $Q$-learning.

**Summary Of The Review:**

I believe that the meta-learning community will find this paper interesting. It provides new insights into formulating meta-learning models and gives examples of such insights being applied to obtain empirical gains. The paper is well-written and features exemplary empirical execution.

POST-REBUTTAL: I have updated my score to 10 and confidence to 5. I think this paper should get an oral, if not best paper. It is the best in my batch and is arguably the best I have read all year. It may be the best paper I have ever peer-reviewed.

---

> ### Author Response · Authors · 2021-11-23
> **Thank you for your review**
>
> Dear reviewer,
>
> Thank you for an insightful review and feedback, we will take this with us as we revise the manuscript. Please see below for answers to your questions.
>
> - *Learned sequence model update rules*. We are sympathetic to the reviewer’s comment; yet there is only so much we can present in a given space constraint. For the simpler experiments, we opted for fixed update rules so that we can visualize their behavior; for larger experiments, SOTA baselines use fixed update rules. We do agree with the reviewer that this is a great avenue for further research.
>
> - *Theoretical results*. That is entirely fair. Even so, we do believe that it is useful to understand on a high level – even if in a simplified setting – what the method does. To make stronger theoretical claims, further structure is required.  We do hope our analysis can serve as inspiration for future theoretical research in more specific settings.
>
> - *Minor comments*. Thank you for spotting this - you are indeed correct on both accounts.

---

> > ### Comment · Reviewer_N6Le · 2021-11-23
> > **Thank you for your response.**
> >
> > - This is fine. Would you mind sparing a sentence or two in the main text to inform the reader that you do not experiment with such models, despite such models technically falling under your purview? I think this would strengthen the work.
> > - Great!
> > - Great!
> >
> > A further question that is not meant to be antagonistic, rather for my own understanding: is there anything, anything at all, that could explain the empirical difference between STACX* and BMG that is not captured within the text? Is there any "secret sauce" that has escaped mention?
> >
> > Also, is there anything in my original review that is inaccurate, even slightly? You need not worry about offending me. This is a selfish request for some feedback on my reviewing.

---

> > > ### Author Response · Authors · 2021-11-23
> > > **No secret sauce**
> > >
> > > To the best of our knowledge, there is no extra secret sauce that drive this performance gain. The two agents are implemented in the exact same code base - the difference between them is *only* a change of the meta-objective, as detailed in the paper. There are no tricks involved in the implementation of the BMG meta-objective, it is implemented *exactly* as described in appendix C.
> > >
> > > It's a good review - we agree with all points that you brought up!

---

> > > > ### Comment · Reviewer_N6Le · 2021-11-23
> > > > **Thank you.**
> > > >
> > > > I will champion this paper. I wish I had written it myself.

---

> > > > > ### Comment · Reviewer_N6Le · 2021-11-23
> > > > > **Another question.**
> > > > >
> > > > > I am convinced that BMG will be better than MG in most tasks we care about and am eager to try it in my own work. However, the no free lunch theorem tells us that there must be some task in which BMG will do worse than MG. What could such a task look like? Would all examples of such a task be pathological? I realize this is a difficult question.

---

> > > > > > ### Comment · Reviewer_N6Le · 2021-11-23
> > > > > > **Yet another question.**
> > > > > >
> > > > > > What is the relationship between BMG and the Lookahead optimizer [A]?
> > > > > >
> > > > > > References
> > > > > > - [A] Zhang et al., Lookahead Optimizer: $k$ steps forward, 1 step back, NeurIPS 2019, https://arxiv.org/pdf/1907.08610.pdf.

---

> > > > > > > ### Author Response · Authors · 2021-11-23
> > > > > > > **Replies to follow-up questions**
> > > > > > >
> > > > > > > Thank you for the raised score and the kind words, we are glad that we could inspire you! Please see below for replies to your follow-up questions:
> > > > > > >
> > > > > > > - *NFL:* recall that BMG is a strict generalization of MG (Eq. 3). Hence, the comparison is less between two distinct methods, but rather between a specific known instance (MG) and the more general class of methods it belongs to. Thus the question you are asking is whether there are problems where the MG instance is best of the class. In principle, there should be problems where this is the case, although we cannot say directly what such a problem might look like. For MG to be the best solution, it must be the case that L2 is the best matching function; a possible reason for this could be noisy or non-stationary data, so that metrics that involve data sampling (like the KL-divergence) become biased or suffer from high variance. Simultaneously, it must be the case that the best target bootstrap is one gradient step on the objective. It is not clear when this would be the case. For L=1 to be best, the problem should suffer from a highly non-linear loss landscape; however, if it does, a gradient descent step is unlikely to be the best update (any form of curvature correction should be an improvement; c.f. Cor. 1).
> > > > > > >
> > > > > > >
> > > > > > > - *Lookahead Optimizer*. Thank you for mentioning this paper. The basic notion of producing a forward-looking target (in this case, under a different update rule) is indeed the same and is in turn inspired by previous works from few-shot learning. The update to the slow weights, i.e. moving in the direction of the target in parameter space, is equivalent to using an L2 matching function. Whereas our work focuses on meta-learning an update rule, giving rise to meta-gradients, this work focuses on a setting where slow and fast weights are different parameters for the learner that are updated differently. To see how BMG relates to this apprach, consider how it can be reduced to the lookahead optimizer: first, suppose that the meta-parameters $w$ in BMG are used to 'initialize $x$' at each step. For the BMG update, set $K$ (the number of updates to backpropagate through) to $0$. Target matching under the squared Euclidean norm then gives $\nabla_w \mu(\tilde{x}, x^{(K)}) = \nabla_w \mu(\tilde{x}, x) = \nabla_w \mu(\tilde{x}, w) = \nabla_w  1/2|| \tilde{x} - w ||_2^2 = -(\tilde{x} - w)$, and so the BMG update (Eq. 2) becomes $\tilde{w} = w + \beta (\tilde{x} - w)$, which is equivalent to the lookahead optimizer if $\tilde{x}$ is $k$ steps under a fixed algorithm.

---

### Official Review · Reviewer_gG1V · 2021-11-02

**Correctness:** 4
**Technical Novelty And Significance:** 4
**Empirical Novelty And Significance:** 4
**Recommendation:** 8
**Confidence:** 3

**Main Review:**

**‌Originality**

The paper is original and novel, proposing a new algorithm to overcome two shortcomings of standard meta-optimization algorithms: curvature mismatch and limited evaluation.

**Quality**

The paper is technically sound, and claims are backed by solid experiments in the reinforcement learning and multi-task meta-learning evaluation setting.

**Clarity**

The paper is clear, however, the algorithm description in section 3 is very abstract. I think the paper would benefit from a running example and a dedicated section and pseudo-code describing the algorithm and how it can be instantiated in different experimental settings.

**Significance**

The work is significant and will benefit the reinforcement and meta-learning community by addressing some of the limitation of the current meta-learning algorithms.

**Limitations**

- The theoretical analysis is limited to noiseless 1-step target updates.
- The experimental evaluation in the multi-task meta-learning setting is limited to only compare with MAML on computer vision applications.

**Questions to Authors**

- Some engineering / handcrafting is still required by the machine learning practitioner to select what "target" the meta-learner is going to optimize, as well as the proper "metric" for the meta-learner to optimize for. Could the authors comment a bit about what heuristics they used when making these decisions? and whether the automation for this process is possible or not?
- What would it take to extend the analysis beyond 1-step noiseless target updates?
- How does the performance of BMG compare to alternative meta-learning algorithms like R2D2, Meta-OPT-net and prototypical networks? Have the authors experimented with other meta-learning benchmarks beyond image classification?

**Minor Typos**

- Abstract: "show that metric" -> "show that a metric"


**Summary Of The Paper:**

‌‌The paper presents a new meta-learning algorithm to address two shortcomings of standard meta-optimization algorithms: curvature (the meta-learner's objective is typically constrained to the same type of geometry as the learner), and limited evaluation (the meta-objective is evaluated only with-in a K-step horizon, ignoring future learning dynamics). The proposed algorithm addresses these two issues by minimizing the distance to a bootstrapped target under a chosen metric. Empirically, the new algorithm  achieved a new state-of-the art for model-free agents on the Atari ALE benchmark and yielded gains in multi-task meta-learning. Theoretically, some guarantees on performance improvements are provided.


**Summary Of The Review:**

‌‌

---

> ### Author Response · Authors · 2021-11-23
> **Thank you for your review**
>
> Dear reviewer,
>
> Thank you for a thorough review and questions that will help us improve the manuscript. Please see below for answers to your questions.
>
> - *Choosing target / distance*. Indeed, in this paper the practitioner must choose the bootstrap and metric. Our goal was to present a strategy that is simple, generally applicable, and that has few hyper-parameters to tune. Thus, the strategy we examine unrolls the meta-learned update rule for a further number of steps $(L-1)$, followed by a gradient step on the objective to ensure the target is grounded. The intuition for this bootstrap comes from acceleration - the target is a future point on the trajectory of the learned update rule. The final gradient step pushes the trajectory in a descent direction. For the matching function, we similarly opted for popular (pseudo-)metrics, L2 and KL, and found that they worked well. With that said, it should be possible to design better bootstraps / metrics in specific settings by exploiting some known structure (e.g. convexity). We believe that deriving principled target bootstraps for specific problem settings is an exciting area for future research.
>
> - *Extending analysis*. As mentioned above, this is an exciting avenue for future research. To provide a more insightful result, we have revamped Thm 1 to more clearly demonstrate the inherent trade-offs in choosing the target bootstrap. To say something sharper about the bootstrap, further structure must be imposed on the problem. Different structures will yield different solutions. For instance, if the objective is assumed to be convex, we can replace the Taylor Series approximations with global inequalities, which makes the analysis amenable to longer bootstraps. Alternatively, if some structure is imposed on the meta-learned update rule, it would be possible to analyze the L>1 setting even if the objective is non-convex.
>
> - *Other comparisons*. As the reviewer noted in their summary, beyond image classification, we report a new state-of-the-art result on a popular meta-learning benchmark, namely Atari (where we double performance relative to the STACX baseline). We have not studied other benchmarks than those reported in the paper. As the purpose of this paper is to provide an improved method for gradient-based meta-learning, we have focused on comparisons to gradient-based meta-learning methods to ensure that our proposed method does indeed offer an improvement among this class of algorithms.

---

### Official Review · Reviewer_u9p2 · 2021-11-02

**Correctness:** 4
**Technical Novelty And Significance:** 4
**Empirical Novelty And Significance:** 4
**Recommendation:** 10
**Confidence:** 4

**Main Review:**

**Originality:** The method proposed in the paper is, to the best of my knowledge, novel. The related work section adequately connects the algorithm with existing work in similar directions.

**Significance:** The empirical results show considerable improvement w.r.t. well-performing baselines; moreover, the general idea behind the method could inspire future research.

**Rigour:** Both the theoretical results and the experimental protocols seem sound and solid to me.

**Strengths**

- The method is based on a conceptually compelling and inspiring idea.
- The empirical results are remarkable. The ablations and additional experiments (also from the Appendix) help in understanding what matters in the practical algorithm, as well as highlighting the important parts of the contribution.
- In particular, being able to meta-learn hyperparameters for a behaviour policy can open up new avenues for exploration in reinforcement learning.
- The theoretical results have a clear scope (although not so large) and provide some easy to understand local improvement guarantees.
- The paper is well-written and generally easy to follow.

**Minor Concerns / Questions**

- I believe the name *matching function* makes the presentation of the method a little bit harder to digest. Since the function is a pseudometric (i.e., the larger it is, the larger the distance from the target), it should really be called with a name that reminds the reader of this nature (e.g., *mismatch function*).
- I enjoyed the theoretical results, but it is a pity that they only deal with targets of specific forms and, especially, with $L=1$ only. Ideally, theoretical result with a dependency on $L$ would shed some light on the benefits and limitations of longer bootstrapping horizons.
- Can the authors elaborate on the connection between the way the bootstrapping target is formed in their method and traditional temporal difference learning? In particular, the grounding role of that subtracted gradient "nudging the trajectory in a descent direction" is the same as the one of the reward in temporal difference learning; but, while the reward is at the beginning of the trajectory, the grounding is here at the end of the optimization subtrajectory. Is there any mathematical connection beyond the general shared motivation?
- As briefly touched upon in some passages of the paper, when the underlying function is highly nonlinear, there is the risk that the bootstrapping mechanism can lead the optimization process in worse areas of the landscape. For instance, if the function in Figure 1 had a bump/plateaux where $\tilde w$ is, the bootstrapping mechanism would cause more troubles than standard meta-gradients. Why is this not happening in practice?

-------
*After rebuttal*: I am happy with the answer the authors provided and the update to the paper, which will help the readers understand the relationship between the proposed method and TD-learning. Overall, the improvements make me believe that this paper should be highlighted at the conference, to give other researchers working in the field the possibility to get inspired by this new idea. I am thus raising my score to 10.

**Summary Of The Paper:**

The paper proposes *Bootstrapped Meta-Learning,* a new meta-learning algorithm for hyperparameter optimization. Drawing inspiration from temporal difference learning techniques in reinforcement learning, the meta-learner is asked to predict the result of additional unrolled steps of the optimization process, by minimizing a carefully selected distance to a target generated during training. This allows for longer meta-learning optimization horizons, without the need for differentiation through longer optimization trajectories. The method is tested for hyperparameter optimization for reinforcement learning, including learning the exploration hyperparameter for a behaviour policy, and in multi-task meta-learning.

**Summary Of The Review:**

The algorithmic and empirical contributions of the paper, as well as the theoretical grounding, largely justify in my opinion its acceptance at the conference.

---

> ### Author Response · Authors · 2021-11-23
> **Thank you for your review**
>
> Dear reviewer,
>
> Thank you for a thoughtful review and excellent questions. We will take your feedback onboard as we prepare a final version of the paper. Please see answers to your questions below.
>
> - *Specific targets and L=1*. We sympathize with the sentiment and have strengthened Thm 1 to be more general (it merges Thm 1. and Cor. 2). In particular, it shows the inherent trade-off between targets that are “safe” and targets that carry high learning signal. In general, without stronger assumptions on the objective, update, and matching function, it is difficult to say something stronger. With that said, we believe that Thm 1 can serve as a useful starting point for more specialized results that develop stronger theoretical insights for specific problem settings.
>
> - *Connection to TD-learning*. The inspiration from TD-learning in our method is the general idea of using future “predictions” as targets. The reviewer raises a good point: in TD-learning, the grounding happens at the beginning of the bootstrap, whereas we ground the target at the end of the bootstrap. The motivation for our approach is that it produces a target on the parameter trajectory of the update rule, thus embodying acceleration (reach future points faster). The final gradient step pushes the target off the trajectory in a descent direction, and as such is guaranteed to be a slightly better target. In contrast, taking a gradient step first and then applying $(L-1)$ bootstrap steps does not guarantee that the target is better than following the update rule for $L-1$ steps. We ran an ablation on Atari with the reviewers suggested TB, but found that it performs worse overall; with L=4, it achieves a median HNS of ~4.8, compared to  ~6.1 for our report result for L=4.
>
> - *Non-linearity and target robustness*. While we generally agree with this point, please note that a short bootstrap (L=1) does not suffer more than the standard meta-gradient (e.g. Eq. 3). Hence this boils down to how “far” it is safe to bootstrap. We observe an inverted U-shape relationship between performance and the length of the bootstrap, depicted in Fig. 2, where increasing the horizon is beneficial up to a point, after which bootstrapping further leads to performance degradation.

---

### Decision · Program_Chairs · 2022-01-20

**Decision:**

Accept (Oral)

**Comment:**

This paper addresses a meta-learning method which involves bilevel optimization. It is claimed that two limitations (myopia of MG and restricted consideration of geometry of search space) that most of existing methods have can be resolved by the MBG with a properly chosen pseudo-metric. The algorithm first bootstraps a target from the meta- learner, then optimizes the meta-learner by minimizing the distance to that target under a chosen pseudo-metric. The authors also establish conditions that guarantee performance improvements and show that metric can be sued to control meta-optimization. All the reviewers agree that the idea is interesting and experiments well support it. Authors did a good job in the rebuttal phase, resolving most of concerns raised by reviewers, leading that two of reviewers raised their score. While the current theoretical results are limited to a simple case where L=1$, the method is attractive for meta-learning community. All reviewers agree to champion this paper. Congratulations on a nice work.